# Enteropathogen antibody dynamics and force of infection among children in low-resource settings

**Benjamin F Arnold[1,2,3]\***, **Diana L Martin[4]**, **Jane Juma[5]**, **Harran Mkocha[6]**, **John B Ochieng[5]**, **Gretchen M Cooley[4]**, **Richard Omore[5]**, **E Brook Goodhew[4]**, **Jamae F Morris[7]**, **Veronica Costantini[8]**, **Jan Vinjé[8]**, **Patrick J Lammie[4,9]**, **Jeffrey W Priest[10]**

[1]Division of Epidemiology and Biostatistics, University of California, Berkeley, Berkeley, United States; [2]Francis I. Proctor Foundation, University of California, San Francisco, San Francisco, United States; [3]Department of Ophthalmology, University of California, San Francisco, San Francisco, United States; [4]Division of Parasitic Diseases and Malaria, United States Centers for Disease Control and Prevention, Atlanta, United States; [5]Kenya Medical Research Institute, Centre for Global Health Research, Kisumu, Kenya; [6]Kongwa Trachoma Project, Kongwa, United Republic of Tanzania; [7]Department of African-American Studies, Georgia State University, Atlanta, United States; [8]Division of Viral Diseases, United States Centers for Disease Control and Prevention, Atlanta, United States; [9]Neglected Tropical Diseases Support Center, Task Force for Global Health, Decatur, United States; [10]Division of Foodborne, Waterborne, and Environmental Diseases, United States Centers for Disease Control and Prevention, Atlanta, United States

**\*For correspondence:**
ben.arnold@ucsf.edu

**Competing interests:** The authors declare that no competing interests exist.

**Abstract** Little is known about enteropathogen seroepidemiology among children in low-resource settings. We measured serological IgG responses to eight enteropathogens (*Giardia intestinalis, Cryptosporidium parvum, Entamoeba histolytica, Salmonella enterica*, enterotoxigenic *Escherichia coli, Vibrio cholerae, Campylobacter jejuni*, norovirus) in cohorts from Haiti, Kenya, and Tanzania. We studied antibody dynamics and force of infection across pathogens and cohorts. Enteropathogens shared common seroepidemiologic features that enabled between-pathogen comparisons of transmission. Overall, exposure was intense: for most pathogens the window of primary infection was <3 years old; for highest transmission pathogens primary infection occurred within the first year. Longitudinal profiles demonstrated significant IgG boosting and waning above seropositivity cutoffs, underscoring the value of longitudinal designs to estimate force of infection. Seroprevalence and force of infection were rank-preserving across pathogens, illustrating the measures provide similar information about transmission heterogeneity. Our findings suggest antibody response can be used to measure population-level transmission of diverse enteropathogens in serologic surveillance.
DOI: https://doi.org/10.7554/eLife.45594.001

## Introduction

A broad set of viral, bacterial, and parasitic enteropathogens are leading causes of the global infectious disease burden, with the highest burden among young children living in lower income countries (*GBD 2016 DALYs and HALE Collaborators, 2016*). Infections that result in acute diarrhea and related child deaths drive disease burden estimates attributed to enteropathogens, but

**eLife digest** Diarrhea, which is caused by bacteria such as *Salmonella* or by viruses like norovirus, is the fourth leading cause of death among children worldwide, with children in low-resource settings being at highest risk. The pathogens that cause diarrhea spread when stool from infected people comes into contact with new hosts, for example, through inadequate sanitation or by drinking contaminated water. Currently, the best way to track these infections is to collect stool samples from people and test them for the presence of the pathogens. Unfortunately, this is costly and difficult to do on a large scale outside of clinical settings, making it hard to track the spread of diarrhea-causing pathogens.

The body produces antibodies – small proteins that can detect specific pathogens – in response to an infection. These antibodies help ward off future infections by the same pathogen, so if they are present in the blood, this indicates a current or previous infection. Scientists already collect blood samples to track malaria, HIV and vaccine-preventable diseases in low-resource settings. These samples could be tested more broadly to measure the levels of antibodies against diarrhea-causing pathogens.

Now, Arnold et al. have used blood samples collected from children in Haiti, Kenya, and Tanzania to measure antibody responses to 8 diarrhea-causing pathogens. The results showed that many children in these settings had been infected with all 8 pathogens before age three, and that all of the pathogens shared similar age-dependent patterns of antibody response. This finding enabled Arnold et al. to combine antibody measurements with statistical models to estimate each pathogen's force of infection, that is, the rate at which susceptible individuals in the population become infected. This is a key step for epidemiologists to understand which pathogens cause the most infections in a population.

The experiments show that testing blood samples for antibodies could provide scientists with a new tool to track the transmission of diarrhea-causing pathogens in low-resource settings. This information could help public health officials design and test efforts to prevent diarrhea, for example, by improving water treatment or developing vaccines.
DOI: https://doi.org/10.7554/eLife.45594.002

asymptomatic infections are extremely common and the full scope of sequelae is only partially understood (*Liu et al., 2016*; *Platts-Mills et al., 2018*). Much of what we know about enteropathogen transmission is based on passive clinical surveillance, which reflects a small fraction of all infections. For example, antibody-based incidence of infection to *Salmonella enterica* and *Campylobacter jejuni* were 2–6 orders of magnitude higher than case-based surveillance in European populations (*Simonsen et al., 2008*; *Falkenhorst et al., 2012*; *Teunis et al., 2012*; *Teunis et al., 2013*), and a study of *Salmonella enterica* serotype Typhi in Fiji found similarly high discordance between antibody-based incidence and case-based surveillance (*Watson et al., 2017*). A more complete picture of enteropathogen infection in populations would help understand drivers of transmission, disease burden, naturally acquired immunoprotection, as well as to design public health prevention measures, and measure intervention effects.

Stool-based, high-throughput PCR assays have helped solve the logistical difficulties of single-pathogen testing for enterics and have provided new insights into pathogen-specific infections and disease burden (*Liu et al., 2016*; *Platts-Mills et al., 2018*). Yet, stool is not routinely collected in population-based surveys, and infection with many globally important enteric pathogens can be sufficiently rare and relatively short-lived to require designs with almost continuous surveillance (*Platts-Mills et al., 2018*; *Lin et al., 2018*). At the same time, large-scale serological surveillance platforms create new opportunities for expanded enteropathogen surveillance alongside other infectious diseases (*Metcalf et al., 2016*; *Arnold et al., 2018*). These challenges and opportunities have generated interest in antibody-based measurement as a complement to PCR for population-based enteropathogen surveillance (*Griffin et al., 2011*; *Exum et al., 2016*; *Moss et al., 2014*; *Arnold et al., 2017*), and for endpoints in observational and randomized studies (*Crump et al., 2007*; *Zambrano et al., 2017*; *Chard et al., 2018*; *Vargas et al., 2017*; *Mosites et al., 2018*; *Wade et al., 2018*; *Egorov et al., 2018*).

After infection, many enteropathogens elicit a transiently elevated antibody response that wanes over time. In lower transmission settings where antibody responses could be monitored longitudinally after distinct infections, *Salmonella enterica*, *Campylobacter jejuni*, *Cryptosporidium parvum*, and *Giardia intestinalis* (syn. *Giardia lamblia*, *Giardia duodenalis*) immunoglobulin G (IgG) levels in blood have been shown to wane over a period of months since infection; IgM and IgA levels decline even more quickly (*Teunis et al., 2012*; *Strid et al., 2001*; *Priest et al., 2001*; *Priest et al., 2010*; *Falkenhorst et al., 2013*; *Hjøllo et al., 2018*). Compared with permanently immunizing infections such as measles, transient immunity adds a layer of complexity to seroepidemiologic inference and methods. To our knowledge, there has been no detailed study of enteropathogen seroepidemiology among children in low-resource settings where transmission is intense beginning early in life (*Platts-Mills et al., 2018*). Such studies are needed to determine if serology is a viable approach to measure enteropathogen transmission in low-resource settings.

We conducted a series of analyses in cohorts from Haiti, Tanzania, and Kenya that measured serological antibody responses to eight enteropathogens using multiplex bead assays. Our objectives were to identify common patterns in antibody dynamics shared across enteropathogens and populations, and to evaluate serological methods to compare between-pathogen heterogeneity in infection, including estimates of force of infection. Our results provide new insights into the seroepidemiology of enteropathogens among children living in low-resource settings, and contribute advances to inform the design and analysis of surveillance efforts whose goal is to quantify heterogeneity in enteropathogen transmission through antibody response.

## Results

### Study populations

The analysis included measurements from cohorts in Haiti, Kenya, and Tanzania. Blood specimens were tested for IgG levels to eight enteropathogens using a multiplex bead assay on the Luminex platform (*Table 1*). The Haitian cohort included repeated measurements among children enrolled in a study of lymphatic filariasis transmission in Leogane from 1990 to 1999 (*Lammie et al., 1998*; *Hamlin et al., 2012*). Leogane is a coastal agricultural community west of Port au Prince. At the time of the study its population was approximately 15,000, most homes had no electricity and none had running water. In total, the Haiti study tested 771 finger prick blood specimens collected from 142 children ages birth to 11 years old, with each measurement typically separated by one year (median measurements per child: 5; range: 2 to 9). In Kenya, a 2013 prospective trial of locally-produced, in-home ceramic water filters enrolled 240 children in a serological substudy (*Morris et al., 2018*). Study participants were identified through the Asembo Health and Demographic Surveillance System, which is located in a rural part of Siaya County, western Kenya along the shore of Lake Victoria. Only 29% of the population had piped drinking water (public taps), water source contamination with *E. coli* prevailed (93% of samples tested), and the average age children began consuming water was 4 months (*Morris et al., 2018*). Children aged 4 to 10 months provided dried blood spot specimens at enrollment (February 2013), and again 6 to 7 months later (August to September, 2013; n = 205 children measured longitudinally). The Kenya study period encompassed seasonally heavy rains from March through June. In Tanzania, 96 independent clusters across eight trachoma-endemic villages in the Kongwa region were enrolled in a randomized trial to study the effects of annual azithromycin distribution on *Chlamydia trachomatis* infection (*Wilson et al., 2019*). The population is very rural, and water is scarce in the region: at enrollment, 69% of participants reported their primary drinking water source, typically an unprotected spring, was >30 min' walk one-way. From 2012 to 2015, the Tanzania study collected dried blood spots from between 902 and 1577 children ages 1–9 years old in annual cross-sectional surveys that took place from October through December at the conclusion of the dry season before heavy seasonal rains (total measurements: 4,989). Although children could have been measured repeatedly over the four-year study in Tanzania, they were not tracked longitudinally. There was no evidence that the Kenya and Tanzania interventions reduced enteropathogen antibody response (*Supplementary file 1*), so this analysis pooled measurements from the study arms in each population.

**Table 1.** Number of children and samples tested, and estimated seropositivity cutoffs by country and antigen included in the seroepidemiologic analyses.

| | N children | N samples | Seropositivity cutoff, $\log_{10}$ IgG (MFI-bg) * | | |
| | | | External Reference | Mixture Model | Presumed Unexposed |
|---|---|---|---|---|---|
| Leogane, Haiti | | | | | |
| *Giardia* VSP-3 | 142 | 771 | 2.42 | 1.64 | 2.11 |
| *Giardia* VSP-5 | 142 | 771 | 2.31 | 1.46 | 1.88 |
| *Cryptosporidium* Cp17 | 142 | 771 | 2.26 | 2.00 | 2.58 |
| *Cryptosporidium* Cp23 | 142 | 771 | 2.70 | 2.75 | 2.57 |
| *E. histolytica* LecA | 142 | 771 | 2.48 | 2.30 | 1.93 |
| *Salmonella* LPS group B | 142 | 771 | | 1.60 | 1.37 |
| *Salmonella* LPS group D | 142 | 771 | | 1.48 | 2.48 |
| ETEC LT B subunit | 142 | 771 | | | 2.86 |
| Norovirus GI.4 | 142 | 771 | | 2.51 | 2.09 |
| Norovirus GII.4.NO | 142 | 771 | | 2.04 | 2.24 |
| Asembo, Kenya | | | | | |
| *Giardia* VSP-3 | 240 | 445 | 2.81 | 1.62 | 1.67 |
| *Giardia* VSP-5 | 240 | 445 | 2.65 | 1.82 | 1.67 |
| *Cryptosporidium* Cp17 | 240 | 445 | 2.63 | 2.58 | 2.38 |
| *Cryptosporidium* Cp23 | 240 | 445 | 3.14 | 3.40 | 2.36 |
| *E. histolytica* LecA | 240 | 445 | | 1.89 | |
| *Salmonella* LPS group B | 240 | 445 | | 1.36 | |
| *Salmonella* LPS group D | 240 | 445 | | 1.41 | |
| ETEC LT B subunit | 240 | 445 | | | 2.79 |
| Cholera toxin B subunit | 240 | 445 | | | 2.91 |
| *Campylobacter* p18 | 240 | 445 | | | 2.11 |
| *Campylobacter* p39 | 240 | 445 | | 2.61 | 2.57 |
| Kongwa, Tanzania | | | | | |
| *Giardia* VSP-3 | 4989 | 4989 | 2.23 | 2.04 | |
| *Giardia* VSP-5 | 4989 | 4989 | 2.15 | 2.27 | |
| *Cryptosporidium* Cp17 | 4989 | 4989 | 2.26 | | |
| *Cryptosporidium* Cp23 | 4989 | 4989 | 2.58 | | |
| *E. histolytica* LecA | 4989 | 4989 | 1.97 | 2.50 | |
| *Salmonella* LPS group B[†] | 902 | 902 | | | |
| *Salmonella* LPS group D[†] | 902 | 902 | | | |
| ETEC LT B subunit | 4989 | 4989 | | | |
| Cholera toxin B subunit[‡] | 4087 | 4087 | | | |
| *Campylobacter* p18[†] | 902 | 902 | | | |
| *Campylobacter* p39[†] | 902 | 902 | | | |

*Seropositivity cutoffs determined using external reference samples (typically ROC curves except for *Giardia* and *E. hystolitica* in Haiti), finite Gaussian mixture models, or distribution among the presumed unexposed (see Materials and methods for details). External reference cutoffs vary across cohorts for the same antigen due to use of different bead sets in each cohort. External reference cutoffs reported from years (2013–2015) in Tanzania, estimated among 4087 samples. Cutoff values are missing if they could not be estimated in each method; cutoff values based on the presumed unexposed required longitudinal measurements within individual children and therefore could not be estimated for any antigen in the repeated cross-sectional design in Tanzania.

[†] Measured only in year 1 of the study (2012).

‡ Measured only in years 2–4 of the study (2013–2015).
DOI: https://doi.org/10.7554/eLife.45594.003

## Age-dependent shifts in population antibody distributions

We estimated seropositivity cutoffs using three approaches: receiver operator characteristic (ROC) curve analyses for *Giardia, Cryptosporidium,* and *Entamoeba histolytica* including a panel of external, known positive and negative specimens, as previously reported (*Moss et al., 2014*; *Morris et al., 2018*); Gaussian mixture models (*Benaglia et al., 2009*) fit to measurements among children ages 0–1 year old to ensure a sufficient number of unexposed children; and, presumed seronegative distributions among children who experienced large increases in antibody levels (*Table 1*). Classification agreement was high between the different approaches (agreement >95% for most comparisons; *Supplementary file 2*).

Among children < 2 years old, antibody levels clearly distinguished seronegative and seropositive subpopulations, but there were not distinct seronegative and seropositive subpopulations by age 3 years for most pathogens measured in Haiti (*Figure 1*) and Tanzania (*Figure 1—figure supplement 1*). By age 3 years, the majority of children were seropositive to *Cryptosporidium*, enterotoxigenic *Escherichia coli* heat labile toxin B subunit (ETEC LT B subunit), and norovirus GI.4 and GII.4; in all cases antibody distributions were shifted above seropositivity thresholds. In contrast, there was a qualitative change in the antibody response distributions to *Giardia, E. histolytica, Salmonella* and

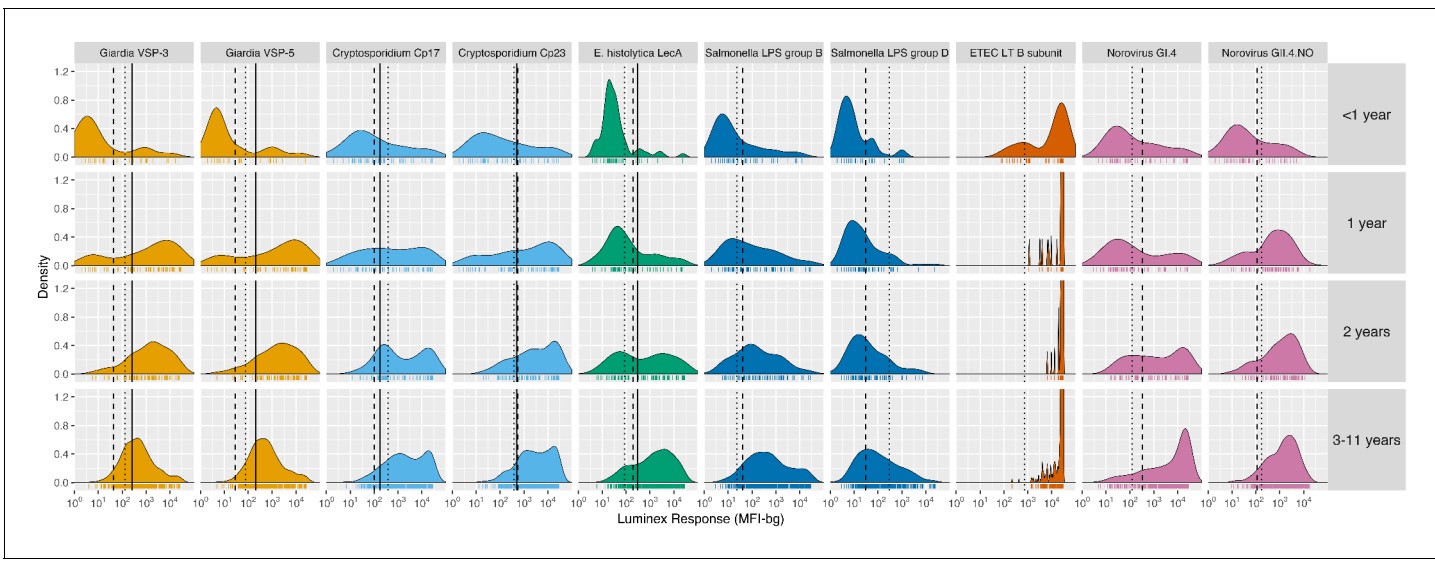

**Figure 1.** Age-stratified, IgG distributions among a longitudinal cohort of 142 children ages birth to 11 years in Leogane, Haiti, 1990 – 1999. IgG response measured in multiplex using median fluorescence units minus background (MFI-bg) on the Luminex platform in 771 specimens, marked with rug plots below each distribution. Vertical lines mark seropositivity cutoffs are based on ROC analyses (solid), finite Gaussian mixture models (heavy dash), or distribution among presumed unexposed (light dash). Mixture models failed to converge for ETEC LT B subunit. Created with notebook (https://osf.io/dk54y) and data (https://osf.io/3nv98). *Figure 1—figure supplement 1* shows similar distributions from the Tanzania study. *Figure 1—figure supplement 2* contrasts *Giardia* VSP-3 distributions with trachoma pgp3 distributions. *Figure 1—figure supplement 3* shows distributions from the Kenyan cohort.

DOI: https://doi.org/10.7554/eLife.45594.004

The following figure supplements are available for figure 1:

**Figure supplement 1.** Age-stratified, IgG distributions among 4989 children ages 1 to 9 years old in Kongwa, Tanzania, 2012–2015.
DOI: https://doi.org/10.7554/eLife.45594.005

**Figure supplement 2.** Contrasting age-dependent changes in distributions of IgG levels to *Giardia* VSP-3 and *Chlamydia trachomatis* pgp3 antigens among 4989 children ages 1 to 9 years old in Kongwa, Tanzania, 2012 – 2015.
DOI: https://doi.org/10.7554/eLife.45594.006

**Figure supplement 3.** IgG distributions among children ages 4 to 17 months old in Asembo, Kenya, 2013.
DOI: https://doi.org/10.7554/eLife.45594.007

*Campylobacter* with increasing age, shifting from a bimodal distribution of seronegative and seropositive groups among children ≤ 1 year old to a unimodal distribution by age 3 years and older (*Figure 1*, *Figure 1—figure supplement 1*). A direct comparison of age-dependent shifts in antibody distributions to *Giardia* VSP-3 antigen and *Chlamydia trachomatis* pgp3 antigen in Tanzania illustrates stark differences in enteropathogen- generated immune responses versus pathogens like *Chlamydia* that elicit a response that consistently differentiates exposed and unexposed subpopulations as children age (*Figure 1—figure supplement 2*). Distributions of IgG levels in the younger Kenyan cohort (ages 4–17 months) showed distinct groups of seropositive and seronegative measurements for most antigens (*Figure 1—figure supplement 3*). IgG responses to ETEC LT B subunit and cholera toxin B subunit were near the maximum of the dynamic range of the assay for nearly all children measured in the three cohorts (*Figure 1*, *Figure 1—figure supplement 1*, *Figure 1—figure supplement 3*), and these IgG levels waned as children aged, presumably from acquired immunity (*Figure 1*, *Figure 1—figure supplement 1*).

## Joint variation in antibody response

We hypothesized that IgG responses to closely related antigens would co-vary but that IgG responses to unrelated antigens would be uncorrelated. Joint variation in individual-level IgG responses aligned with hypothesized relationships based on antigenic overlap and shared epitopes. Responses to *Giardia* VSP antigens were strongly correlated in Haiti (Spearman rank: $\rho$=0.99), Kenya ($\rho$=0.84) and Tanzania ($\rho$=0.97), as would be expected for antigens with conserved conformational epitopes (*Figure 2*). *Cryptosporidium* (Cp17, Cp23) and *Campylobacter* (p18, p39) antigens were strongly correlated, but high within-individual variability suggests that measuring responses to multiple unique recombinant protein antigens yields more information about infection than measuring responses to one alone (*Figure 2*). High correlation between *Salmonella* LPS Groups B and D, between norovirus GI.4 and GII.4, and between ETEC and *V. cholerae* likely reflected antibody cross-reactivity. Correlation could also result from multiple previous infections with different *Salmonella* serogroups or different norovirus genogroups. A comparison across all antigens revealed no other combinations with high correlation (*Supplementary file 3*).

We excluded cholera toxin B subunit antibody responses from remaining analyses because of the difficulty of interpreting its epidemiologic measures in light of high levels of cross-reactivity with ETEC LT B subunit. Heat labile toxin-producing ETEC is very common among children in low-resource settings (*Platts-Mills et al., 2018*), and there was no documented transmission of cholera in the study populations during measurement periods.

## Birth to three years of age: a key window of antibody acquisition and primary seroconversion

For most pathogens, mean IgG levels and seroprevalence rose quickly and plateaued by ages 1 to 3 years in Haiti (*Figure 3*), Kenya (*Figure 3—figure supplement 1*), and Tanzania (*Figure 3—figure supplement 2*). Despite enormous individual-level variation, age-dependent mean IgG curves exhibited characteristic shapes seen across diverse pathogens, and reflected high levels of early-life exposure (*Arnold et al., 2017*). In Haiti, seroprevalence ranged from 66% (*E. histolytica*) to 100% (ETEC LT B subunit) by age 3 years (*Figure 3B*), and in Tanzania, the majority of 1 year olds were already seropositive for *Giardia* (77%) and *Cryptosporidium* (85%) (*Figure 3—figure supplement 2B*). There was some evidence of maternally-derived IgG among children under 6 months old with a drop in mean IgG levels by age, but this pattern was only evident for norovirus GI.4 in Haiti (*Figure 3A*) and *Cryptosporidium* Cp17 and Cp23 in Kenya (*Figure 3—figure supplement 1A*). Age-dependent mean IgG responses to many pathogens declined from a young age, presumably from exposure early in life and acquired immunity. Mean IgG levels declined after age 1 year for *Giardia* in Haiti (*Figure 3A*), *Giardia* and *Campylobacter* in Tanzania (*Figure 3—figure supplement 2A*).

## Longitudinal antibody dynamics show significant boosting and waning above seropositivity cutoffs

Based on age-dependent shifts in IgG distributions (*Figure 1*), we hypothesized that conversion of IgG levels to seropositive and seronegative status could mask important dynamics of enteropathogen immune response above seropositivity cutoffs, particularly among ages beyond the window of

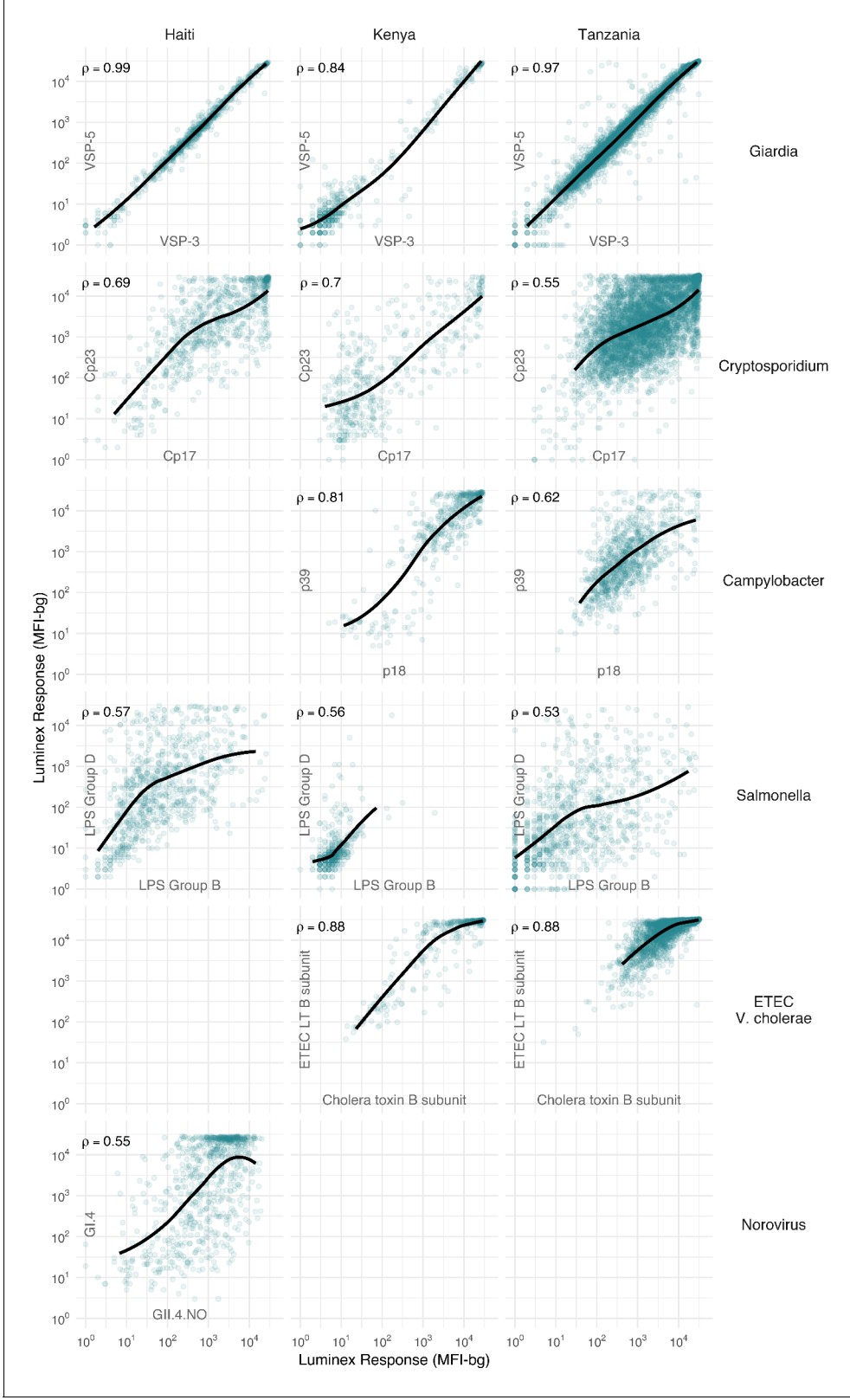

**Figure 2.** Joint distributions of select enteric pathogen antibody responses among children in three cohorts from Haiti, Kenya, and Tanzania. Each panel includes Spearman rank correlations (ρ) and locally weighted regression smoothers with default parameters, trimmed to 95% of the data to avoid edge effects. Antibody response measured in multiplex using median fluorescence units minus background (MFI-bg) on the Luminex platform. Empty
*Figure 2 continued on next page*

*Figure 2 continued*

panels indicate that the antibodies were not measured in that cohort. **Supplementary file 3** includes all pairwise comparisons. Created with notebook (https://osf.io/hv9ce) and data (https://osf.io/3nv98, https://osf.io/2q7zg, https://osf.io/kv4d3).
DOI: https://doi.org/10.7554/eLife.45594.008

primary infection. In Haiti and Kenya we examined longitudinal IgG profiles among children. In the Haitian cohort, which was followed beyond the window of primary infection, children commonly had >4 fold increases and decreases in IgG while remaining above seropositivity cutoffs—a pattern observed across pathogens but particularly clear for *Cryptosporidium* (**Figure 4**). In Kenya, 4-fold increases in IgG largely coincided with a change in status from seronegative to seropositive, presumably because increases in IgG followed primary infection in the young cohort ages 4–17 months (**Figure 4—figure supplement 1**). Many Kenyan children exhibited >4 fold increases and decreases in IgG response to *Campylobacter* p18 and p39 antigens above the seropositivity cutoff, a result of earlier primary infection and/or additional infection and boosting during the study period (**Figure 4—figure supplement 1**).

## Comparison of serology with stool-based measures of infection

The Kenya study monitored diarrhea symptoms in weekly visits between enrollment and follow-up. The study collected stool from children whose caregivers reported diarrhea symptoms in the past 7 days, and tested stool for *Cryptosporidium* and *Giardia* infections using an immunoassay with additional PCR testing for *Cryptosporidium* as previously described (**Morris et al., 2018**). Among 132 children with paired serology measurements, those infected with *Cryptosporidium* (n = 17) and *Giardia* (n = 25) enabled us to compare stool-based measures of infection with IgG responses. Children with confirmed infections in diarrheal stools had higher IgG levels and seroprevalence at both time points compared with those who did not have confirmed infections, but many children without stool-confirmed infections seroconverted during the study. Among children without confirmed *Giardia* infection in diarrheal stools, seroprevalence to VSP-3 or VSP-5 antigens increased from 1% (95% CI: 0%, 5%) at enrollment to 22% (14%, 32%) at follow-up; among children without confirmed *Cryptosporidium* infection, seroprevalence to Cp17 or Cp23 antigens increased from 16% (10%, 24%) at enrollment to 47% (37%, 57%) at follow-up. These findings suggest that many children were not shedding genetic material at the time of diarrheal stool collection, or many infections with these two pathogens were asymptomatic. **Supplementary file 4** includes additional details.

## Serological estimates of force of infection

The seroconversion rate, an instantaneous rate of seroconversion among those who are susceptible, is one estimate of a pathogen's force of infection and a fundamental epidemiologic measure of transmission (**Hens et al., 2012**). Serologically derived force of infection is useful for pathogens that commonly present asymptomatically, such as many enteric infections. Across diverse pathogens, steeper age-seroprevalence curves typically reflect higher transmission intensity (**Corran et al., 2007**; **Pinsent et al., 2018**), and age-adjusted seroprevalence equals the area under the age-seroprevalence curve (a summary measure) (**Arnold et al., 2017**). We therefore hypothesized that seroprevalence and prospectively estimated force of infection should embed similar information about infection heterogeneity across pathogens. We also hypothesized that standard methods to estimate force of infection from age-structured seroprevalence would underestimate force of infection derived from longitudinal data because of significant antibody boosting and waning above seropositivity cutoffs.

Longitudinal designs in Haiti and Kenya enabled us to use individual child antibody profiles to estimate average rates of prospective seroconversion and seroreversion during the studies. We defined incident seroconversions and seroreversions as a change in IgG across a pathogen's seropositivity cutoff and estimated force of infection as incident changes in serostatus divided by person-time at risk. In a secondary analysis, we defined incident boosting as a $\geq 4$ fold increase in IgG to a final level above a seropositivity cutoff and incident waning as $\geq 4$ fold decrease in IgG from an initial level above a seropositivity cutoff. The secondary definition captured large changes in IgG

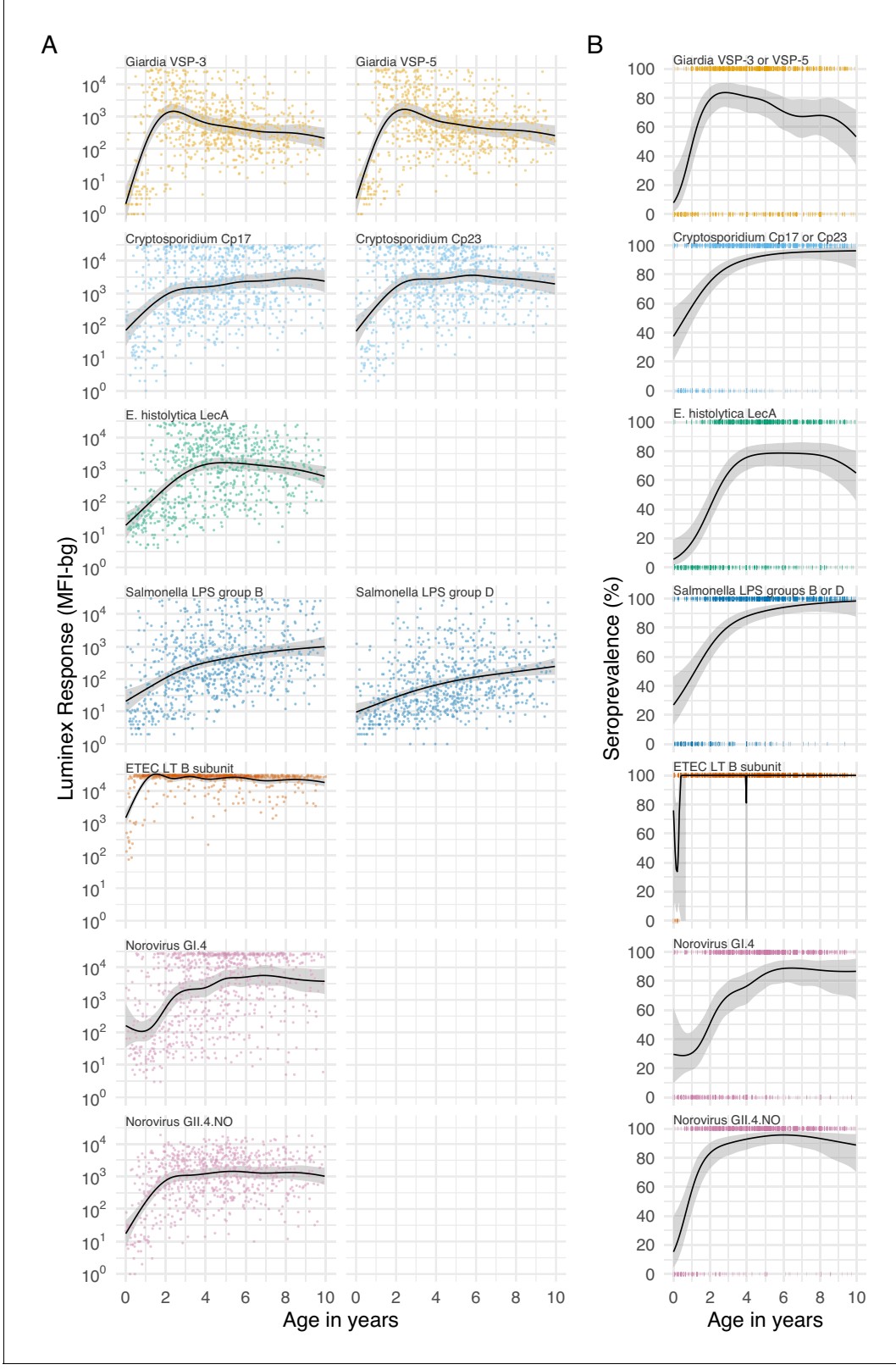

**Figure 3.** Age dependent mean IgG and seroprevalence in Haiti. Geometric means (**A**) and seroprevalence (**B**), estimated with cubic splines among children ages birth to 10 years in Leogane, Haiti 1990–1999. Shaded bands are approximate, simultaneous 95% confidence intervals. IgG response measured in multiplex using median fluorescence units minus background (MFI-bg) on the Luminex platform (N = 771 measurements from 142 children). Created with notebook (https://osf.io/jeby3) and data (https://osf.io/3nv98). Data for some antigens measured among children < 5 years

*Figure 3 continued on next page*

*Figure 3 continued*

previously published (*Arnold et al., 2017*). *Figure 3—figure supplement 1* includes similar curves from Kenya; *Figure 3—figure supplement 2* includes similar curves from Tanzania.

DOI: https://doi.org/10.7554/eLife.45594.009

The following figure supplements are available for figure 3:

**Figure supplement 1.** Age dependent mean IgG and seroprevalence in Kenya.

DOI: https://doi.org/10.7554/eLife.45594.010

**Figure supplement 2.** Age dependent mean IgG and seroprevalence in Tanzania.

DOI: https://doi.org/10.7554/eLife.45594.011

above seropositivity cutoffs, which aligned with repeated boosting and waning observed in the Haitian cohort (*Figure 4*).

We found a rank-preserving relationship between pathogen seroprevalence and average force of infection in Kenya and Haiti (*Figure 5*). Overall levels and steepness of the relationship differed between cohorts, presumably because Kenya measurements were within a window of primary infection for most children (4–17 months) whereas Haiti measurements extended from birth to 11 years and captured lower incidence periods with overall higher seroprevalence as children aged. Consistent with this interpretation, when we progressively narrowed the age range of the Haitian cohort and repeated the analysis, the relationship was steeper when estimated among children ages 0–2 years and flattened as measurements among older children were added (*Figure 5—figure supplement 1*).

Force of infection varied widely across pathogens in Kenya, ranging from 0.1 seroconversions per year for *E. histolytica* to >5 for *Campylobacter* (*Figure 5*). In Haiti, force of infection ranged from 0.3 *E. histolytica* seroconversions per year to 1.1 ETEC seroconversions per year (*Figure 5*). Force of infection estimated from 4-fold changes in IgG led to more events and slightly higher rates compared with those estimated from seroconversion alone (*Table 2*). For example, *Cryptosporidium* incident cases increased from 70 to 204 (a 2.9 fold increase) and the average rate increased from 0.6 (95% CI: 0.5, 0.8) to 0.9 (95% CI: 0.7, 1.0) per child-year when using a 4-fold IgG change criteria because of substantial IgG boosting and waning above the seropositivity cutoff (*Figure 4*). Sensitivity analyses that defined incident boosting over a range of 2-fold to 10-fold increases in IgG showed force of infection estimates were relatively stable across a wide range of definitions. In Haiti, the only pathogen for which force of infection estimated using a 4-fold increase in IgG was significantly higher than the seroconversion rate was *Cryptosporidium* (*Supplementary file 5*).

We evaluated whether model-based force of infection estimates from age-structured seroprevalence could accurately recover estimates from the longitudinal analyses. We focused on the Kenya cohort since children were measured repeatedly during the ages of primary infection and because longitudinal force of infection and seroreversion rate estimates varied considerably across pathogens (*Figure 5*). We estimated force of infection from seroprevalence curves using methods developed for cross-sectional, 'current status' data, a common approach in serosurveillance of vaccine preventable diseases (*Hens et al., 2012*), malaria (*Corran et al., 2007*), and dengue (*Ferguson et al., 1999*; *Katzelnick et al., 2018*). Force of infection estimates from semiparametric spline models were similar to estimates from the longitudinal analysis for all pathogens, but had substantially wider confidence intervals owing to the loss of information from ignoring the longitudinal data structure (*Figure 6*). Parametric approaches including an exponential survival model (*Jewell and Laan, 1995*) and a reversible catalytic model (*Corran et al., 2007*) yielded narrower confidence intervals than the semiparametric model but tended to underestimate force of infection compared with longitudinal estimates (*Figure 6*). Across pathogens, model-based force of infection estimates derived from seroprevalence were rank-preserving compared with nonparametric longitudinal analyses.

We conducted a simulation study to investigate whether longer sampling intervals in the cohorts (6 months in Kenya, 12 months in Haiti) could lead us to miss more frequent exposures and thus under-estimate force of infection. For each cohort, we created 100 imputed datasets that reconstructed a child's daily IgG levels, assuming that each infection boosted IgG and that it would wane exponentially. The simulation drew IgG boosts from empirical distributions in each cohort and used antibody-specific decay rates. We allowed for the maximum number of intermediate exposures between measurements as long as IgG levels could wane sufficiently to follow a child's empirical

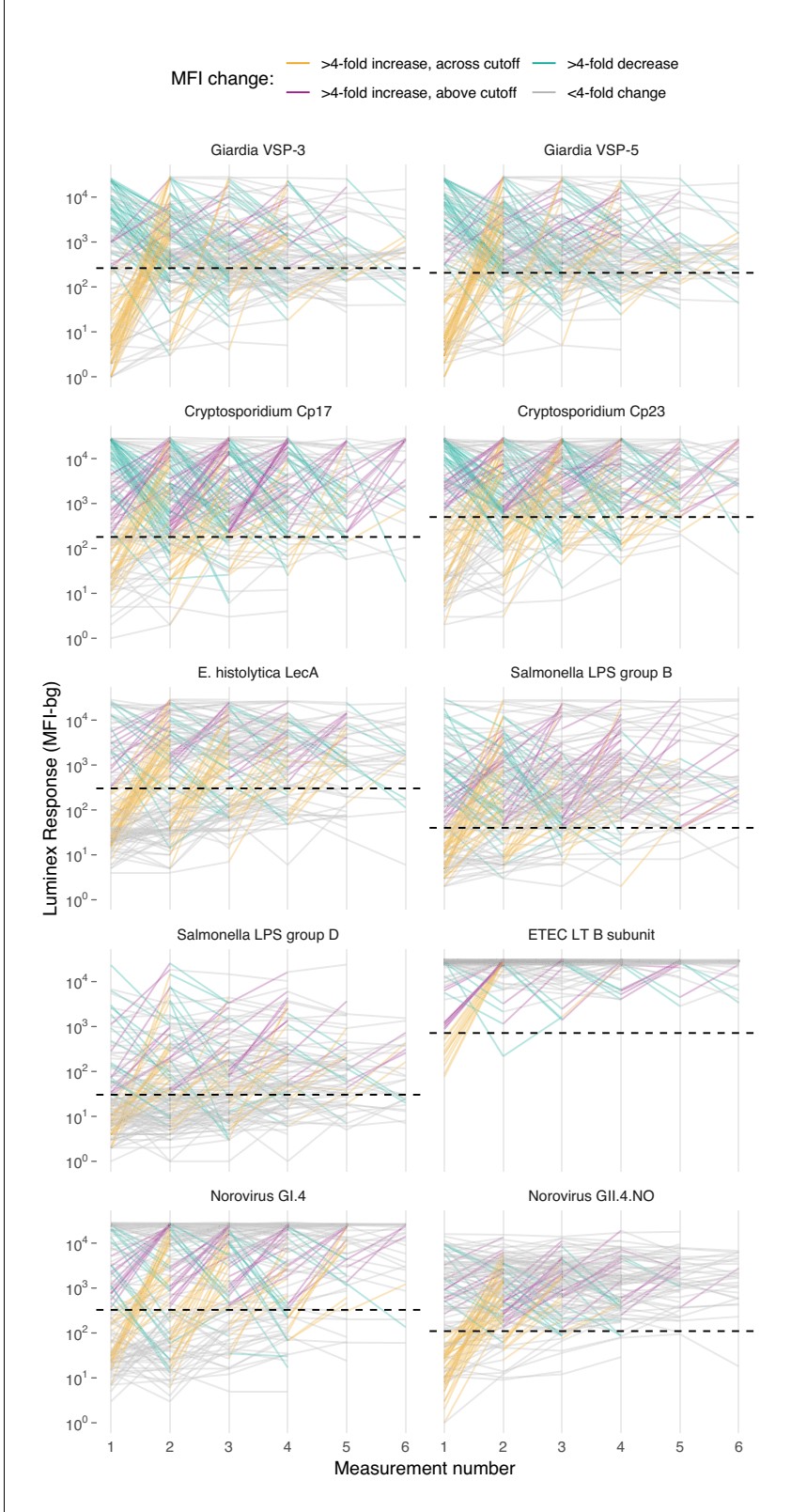

**Figure 4.** Longitudinal changes in IgG response over six repeated measurements among 142 children ages 0–11 years in Leogane, Haiti, 1990 – 1999. Measurements were spaced by approximately 1 year (median spacing = 1, IQR = 0.7, 1.3). Horizontal dashed lines mark seropositivity cutoffs for each antibody. The number of children measured at each visit was: $n_1$ = 142, $n_2$ = 142, $n_3$ = 140, $n_4$ = 131, $n_5$ = 111, $n_6$ = 66); 29 children had >6 measurements that are not shown. IgG response measured in multiplex using median fluorescence units minus background (MFI-bg) on the Luminex

*Figure 4 continued on next page*

*Figure 4 continued*

platform. Created with notebook (https://osf.io/vyhra), which includes additional visualizations, and data (https://osf.io/3nv98). *Figure 4—figure supplement 1* includes a similar figure from the Kenya cohort. *Figure 4—figure supplements 2* and *3* summarize the proportion of children in each category across measurement rounds in Haiti and Kenya.

DOI: https://doi.org/10.7554/eLife.45594.012

The following figure supplements are available for figure 4:

**Figure supplement 1.** Longitudinal changes in IgG response between enrollment and follow-up among 205 children ages 4–17 months in Asembo, Kenya, 2013.

DOI: https://doi.org/10.7554/eLife.45594.013

**Figure supplement 2.** Proportion of children ages 0–11 years with different longitudinal changes in IgG response over six repeated measurements in Leogane, Haiti, 1990 – 1999.

DOI: https://doi.org/10.7554/eLife.45594.014

**Figure supplement 3.** Proportion of children ages 4–17 months with different longitudinal changes in IgG response between enrollment and follow-up 6 months later in Asembo, Kenya, 2013.

DOI: https://doi.org/10.7554/eLife.45594.015

measurements, thus providing an approximate upper bound of the seroconversion rates (force of infection) that could plausibly be detected for each antibody. We down-sampled the daily datasets at intervals of 30, 90, 180, and 360 days to reflect realistic measurement intervals and estimated seroconversion rates. We found that for most pathogens studied, higher resolution sampling would not substantially increase seroconversion rates in absolute terms. Rates estimated through simulation increased from between 0.1 to 0.9 episodes per child-year at risk if measured with a sampling interval of 30 days instead of annually (Haiti) or every six months (Kenya). However, for pathogens with highest seroconversion rates, ETEC and *Campylobacter*, increases in rates estimated with 30 day sampling intervals detected a median of 4 to 8 additional seroconversions per child-year at risk compared with empirical rates (*Figure 7*). In relative terms, seroconversion rates in Haiti had a larger discrepancy; rates more than doubled when using a 30 day sampling interval, compared with the annual interval used in the study.

## Discussion

In cohorts from Haiti, Kenya, and Tanzania we identified consistent patterns in IgG responses that provide new insights into enteropathogen seroepidemiology among children in low-resource settings. Most population-level heterogeneity in IgG levels and seroconversion was between birth and 3 years, reflecting high transmission and early life primary infection. For particularly high transmission pathogens (e.g., ETEC, *Campylobacter*), most variation in IgG levels was observed among children < 1 year old. In these study populations, endemic force of infection for enteropathogens was as high, and in many cases several fold higher, than force of infection estimated during epidemics of new dengue serotype introductions in Nicaragua and Peru (approximately 0.4 to 0.6 seroconversions per year) (*Katzelnick et al., 2018*; *Reiner et al., 2014*). Significant boosting and waning of antibody levels above seropositivity cutoffs identified through longitudinal profiles in Haiti and Kenya reinforce the value of longitudinal designs to derive antibody distributions among unexposed and to estimate force of infection.

The shift of IgG distributions from bimodal to unimodal for many pathogens (*Giardia, Cryptosporidium, E. histolytica*, and *Campylobacter*), resulting from a combination of antibody boosting, waning and acquired immunity, complicates the interpretation of seroprevalence at older ages: among older children a seronegative response could either mean the children were never exposed or they were previously exposed but antibody levels waned below seropositivity cutoffs. The age-dependent shift contrasts with more stable differentiation of seronegative and seropositive groups observed for some other antibody responses (e.g., *C. trachomatis* pgp3 in *Figure 1—figure supplement 2*) and likely results from less robust and sustained IgG response following infection. Estimates of IgG half-life in the Haiti cohort were on the order of 10 weeks for most pathogens (*Supplementary file 8*), implying that time to seroreversion would be approximately 1 year without additional exposure (assuming exponential decay $\lambda = 0.01$ corresponding to a 10 week half-life, a starting level of 10,000 MFI, and seropositivity cutoff of 300 MFI; $t = -\log(N_t/N_0)/\lambda = 350$ days). Seroreversion is therefore

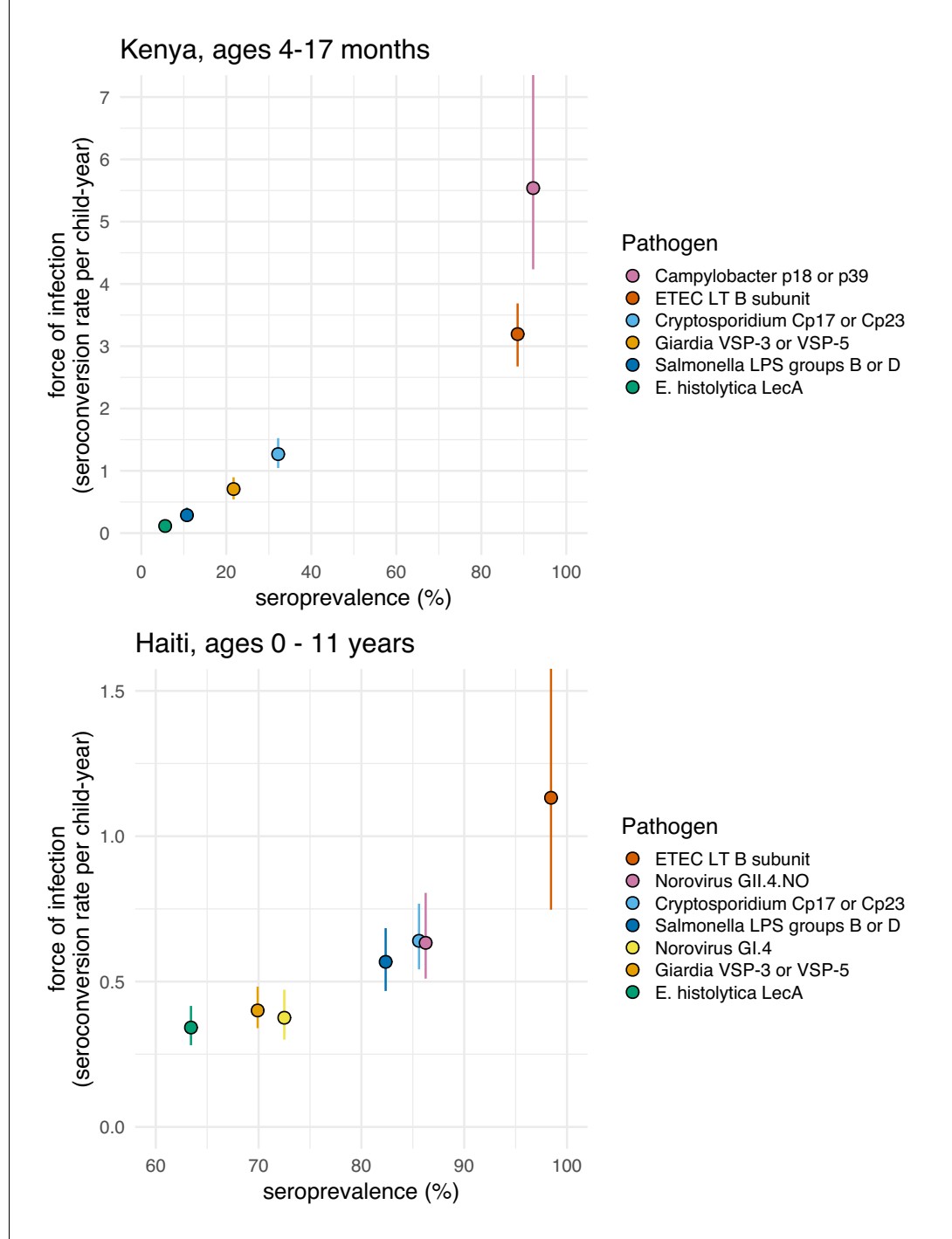

**Figure 5.** Average force of infection versus seroprevalence for enteropathogens measured in the Kenya and Haiti cohorts. Force of infection estimated from prospective seroconversion rates. Vertical lines indicate 95% confidence intervals. Created with notebook (https://osf.io/jp9kf) and data (https://osf.io/2q7zg, https://osf.io/3nv98). *Figure 5—figure supplement 1* includes estimates from Haiti stratified by age bands.
DOI: https://doi.org/10.7554/eLife.45594.016

The following figure supplement is available for figure 5:

**Figure supplement 1.** Average force of infection versus seroprevalence for enteropathogens measured in the Haiti cohort, stratified by different age bands.
DOI: https://doi.org/10.7554/eLife.45594.017

**Table 2.** Incidence rates of seroconversion and seroreversion per child year among children ages 0–11 years in Haiti, 1990–1999.

| Pathogen | Seropositivity cutoff * | | | 4-Fold change in IgG levels † | | | Ratio of cases | Ratio of rates |
|---|---|---|---|---|---|---|---|---|
| | Child-years | Incident cases | Rate (95% CI) | Child-years | Incident cases | Rate (95% CI) | | |
| Seroconversion/boosting | | | | | | | | |
| *Giardia* VSP-3 or VSP-5 | 269.6 | 108 | 0.40 (0.34, 0.48) | 277.2 | 120 | 0.43 (0.35, 0.54) | 1.1 | 1.1 |
| *Cryptosporidium* Cp17 or Cp23 | 109.3 | 70 | 0.64 (0.54, 0.77) | 241.0 | 204 | 0.85 (0.73, 0.97) | 2.9 | 1.3 |
| *E. histolytica* LecA | 283.7 | 97 | 0.34 (0.28, 0.42) | 297.1 | 107 | 0.36 (0.29, 0.45) | 1.1 | 1.1 |
| *Salmonella* LPS groups B or D | 132.1 | 75 | 0.57 (0.47, 0.68) | 226.8 | 149 | 0.66 (0.54, 0.80) | 2.0 | 1.2 |
| ETEC LT B subunit | 9.7 | 11 | 1.13 (0.75, 1.82) | 32.1 | 32 | 1.00 (0.70, 1.45) | 2.9 | 0.9 |
| Norovirus GI.4 | 213.0 | 80 | 0.38 (0.30, 0.47) | 254.3 | 107 | 0.42 (0.34, 0.53) | 1.3 | 1.1 |
| Norovirus GII.4.NO | 105.8 | 67 | 0.63 (0.51, 0.80) | 147.2 | 100 | 0.68 (0.54, 0.86) | 1.5 | 1.1 |
| Seroreversion/waning | | | | | | | | |
| *Giardia* VSP-3 or VSP-5 | 441.6 | 91 | 0.21 (0.17, 0.25) | 290.9 | 127 | 0.44 (0.35, 0.54) | 1.4 | 2.1 |
| *Cryptosporidium* Cp17 or Cp23 | 586.1 | 29 | 0.05 (0.03, 0.07) | 273.5 | 171 | 0.63 (0.53, 0.74) | 5.9 | 12.6 |
| *E. histolytica* LecA | 395.2 | 43 | 0.11 (0.08, 0.15) | 310.4 | 67 | 0.22 (0.16, 0.27) | 1.6 | 2.0 |
| *Salmonella* LPS groups B or D | 544.3 | 25 | 0.05 (0.03, 0.07) | 344.5 | 89 | 0.26 (0.20, 0.32) | 3.6 | 5.6 |
| ETEC LT B subunit | 702.1 | 2 | 0.00 (0.00, 0.01) | 649.7 | 22 | 0.03 (0.02, 0.05) | 11.0 | 11.9 |
| Norovirus GI.4 | 464.9 | 28 | 0.06 (0.03, 0.09) | 362.2 | 56 | 0.15 (0.11, 0.21) | 2.0 | 2.6 |
| Norovirus GII.4.NO | 574.3 | 19 | 0.03 (0.02, 0.05) | 477.9 | 39 | 0.08 (0.06, 0.11) | 2.1 | 2.5 |

*Incident changes in serostatus defined by crossing seropositivity cutoffs.

†Incident changes in serostatus defined by a 4-fold increase or decrease in IgG levels (MFI-bg), with incident boosting episodes restricted to changes that ended above the seropositivity cutoff and incident waning episodes restricted to changes that started from above the seropositivity cutoff.

DOI: https://doi.org/10.7554/eLife.45594.018

possible at young ages, but population-level mean IgG levels and seroprevalence declined for only three pathogens, *Giardia*, ETEC, and *Campylobacter*, and only above age 3 years (*Figure 3*, *Figure 3—figure supplement 2*). Increases in mean IgG levels and seroprevalence with age imply IgG boosting from new infections or repeated infections outpaced IgG decay for all enteropathogens studied until at least age 3 years, and for many pathogens through age 10 years; seroprevalence thus reflects a conservative lower bound of a population's cumulative exposure over this age range.

The age range over which seroprevalence provided useful epidemiologic information varied by pathogen and cohort. In Haiti, 100% of children were seropositive to ETEC LT B toxin before age 12 months, though seroprevalence did not exceed 90% for most other pathogens until age 5 years in Haiti (*Figure 3*). In Kenya, the age range of 4–18 months captured wide variation in seroprevalence for most pathogens. The Tanzania study enrolled children ages one and older due to a primary focus on trachoma monitoring, but missed the key window of variation in antibody response for all enteropathogens except *E. histolytica* and *Salmonella* (*Figure 3—figure supplements 1* and *2*). Studies that extend beyond 10 years into adolescence and adulthood would help determine whether enteropathogen seroprevalence remains sufficiently high that it no longer provides useful epidemiologic information. The shift in IgG distributions for some enteropathogens raises the question of whether population mean IgG levels stabilize at a new 'set point' with repeated infections as has been observed for dengue serotypes (*Salje et al., 2018*); if so, then the use of fold-changes in IgG would be preferred to seropositivity cutoffs to identify incident infections among older ages.

In low-resource settings, measuring a sufficient number of young children before primary infection, preferably with longitudinal measurements, will help ensure that within-sample seropositivity cutoff estimation is possible. Two-component mixture models fit the data and provided reasonable cutoff estimates only when restricted to an age range that included clearly delineated subpopulations of seronegative and seropositive responses. For most pathogens studied, this required measurements among children < 1 year old, an age range during which IgG responses still followed a

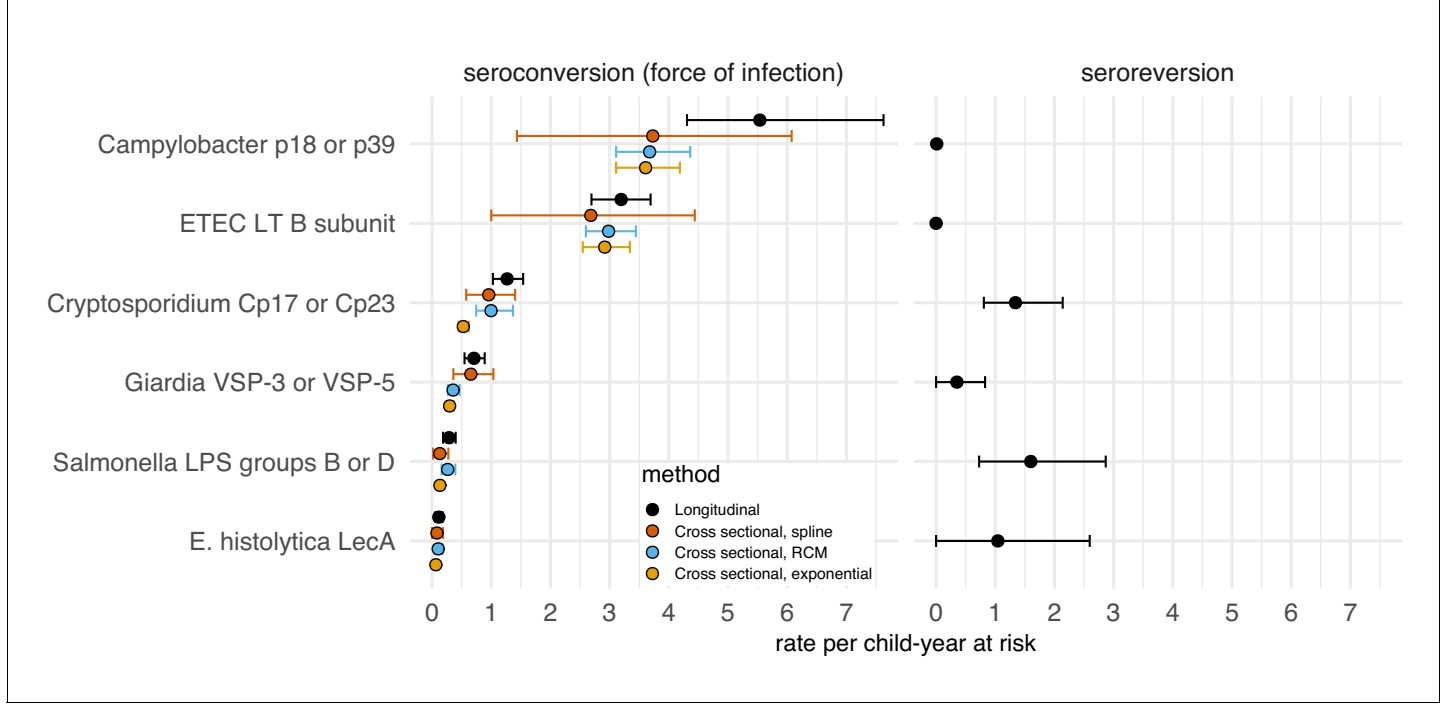

**Figure 6.** Enteropathogen seroconversion and seroreversion rates among 205 children ages 4 to 17 months measured longitudinally in Asembo, Kenya, 2013. The seroconversion rate is a measure of a pathogen's force of infection. Longitudinal estimates are non-parametric rates of incident seroconversions and seroreversions among children at risk, assumed to occur at the midpoint of the measurement interval. Cross-sectional estimators were derived from age-specific seroprevalence curves using semiparametric cubic splines (spline), a reversible catalytic model (RCM) that assumed constant seroconversion and seroreversion rates with the seroreversion rate estimated from prospective data, and a parametric constant rate survival model (exponential). Error bars mark 95% confidence intervals. IgG response measured in multiplex using median fluorescence units minus background (MFI-bg) on the Luminex platform (N = 410 measurements from 205 children). Created with notebooks (https://osf.io/sqvj7, https://osf.io/j9nh3) and data (https://osf.io/2q7zg).

DOI: https://doi.org/10.7554/eLife.45594.019

bimodal distribution. The only reliable approach to estimate seropositivity cutoffs for the highest transmission pathogens like ETEC and *Campylobacter* was to estimate a distribution among presumed unexposed by identifying measurements among children who subsequently experienced a large increase in IgG, a strategy only possible in a longitudinal design. Although not attempted here, studies that wish to compare antibody response and seroprevalence between different sites should use a common assay platform and materials (e.g., shared bead coupling) with jointly estimated seropositivity cutoffs to help ensure comparability across sites, since there are currently no global reference standards to translate arbitrary units into antibody titers for enteropathogens.

Seroepidemiologic measures that can be estimated from cross-sectional surveys are of particular interest for infectious diseases because most large-scale, population-based serosurveillance platforms use cross-sectional designs (*Arnold et al., 2018*). Our results show that seroprevalence and force of infection estimated from seroprevalence models adequately summarize between-pathogen heterogeneity in transmission when compared with longitudinal estimates of force of infection. Pathogens with fastest rising mean IgG and seroprevalence with age (e.g., ETEC, *Campylobacter*, norovirus GII.4; *Figure 3*, *Figure 3—figure supplement 1*) had highest force of infection measured prospectively over the study period, and seroprevalence was rank-preserving with prospective force of infection in both Haiti and Kenya (*Figure 5*). Seroprevalence alone thus appears to be sufficient to assess relative pathogen transmission if measured in an age range that captures ample heterogeneity in response (in these cohorts < 3 years old). Our findings align with modeling studies of other infectious diseases such as malaria (*Corran et al., 2007*), trachoma (*Pinsent et al., 2018*), and dengue (*Katzelnick et al., 2018*), and suggest that enteropathogens share similar seroepidemiologic features conducive to population-based surveillance in cross-sectional surveys despite different

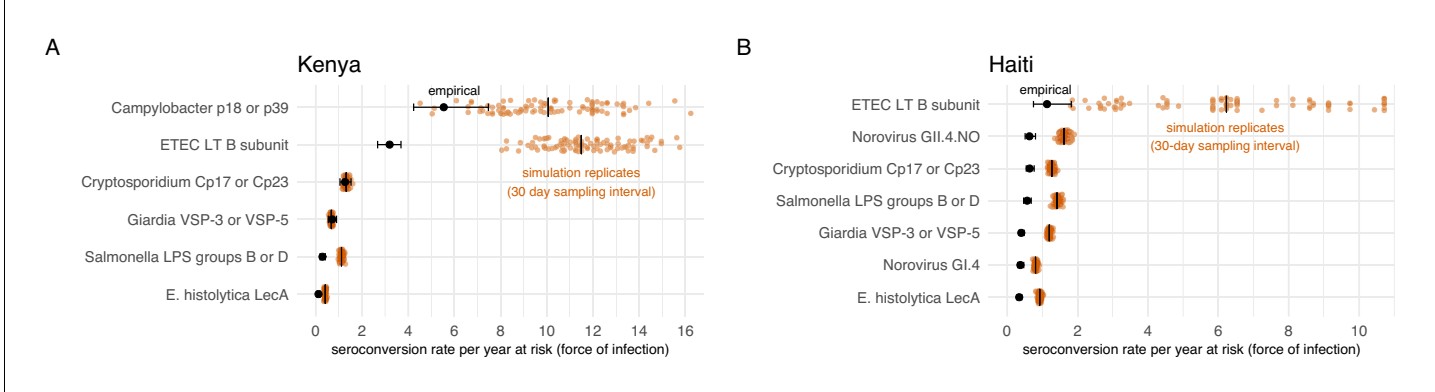

**Figure 7.** Empirical seroconversion rates compared with estimates from 100 simulated datasets with daily resolution IgG trajectories that were sampled at a 30 day interval before estimating seroconversion rates. Vertical lines in the simulation results indicate medians. (**A**) In the Kenya cohort, children were ages 4–18 months and empirical IgG measurements were measured every 6 months (approximately 180 days). (**B**) In the Haiti cohort, children were ages 0–11 years, and empirical IgG measurements were measured approximately each year (median spacing = 1 year, IQR = 0.7, 1.3). Created with notebooks (https://osf.io/qmdf2, https://osf.io/9fxhb) and data (https://osf.io/2q7zg, https://osf.io/3nv98).

DOI: https://doi.org/10.7554/eLife.45594.020

underlying immunology. Two caveats are that cross-sectional methods considered in this study will generally underestimate force of infection because they fail to rigorously account for seroreversion and incident boosting above seropositivity cutoffs (*Figure 6*), and seroprevalence is not a clear marker of cumulative exposure as children age beyond the window of primary infection because of waning IgG responses.

The relative intensity of pathogen transmission based on serological measures in this analysis aligns with relative differences in infection from large-scale molecular testing in stool. For example, rapid rises in IgG levels, seroprevalence, and force of infection for ETEC and *Campylobacter* beginning very early in life is consistent with their prominence in molecular testing of stool specimens in the MAL-ED studies (*Platts-Mills et al., 2018*; *Platts-Mills et al., 2015*). Higher force of infection for norovirus GII.4 compared with norovirus GI.4 in Haiti is consistent with the relative prominence of circulating genogroups (*Bányai et al., 2018*; *Rouhani et al., 2016*). The results lend additional support to the idea that broader serological testing in existing serosurveillance platforms represents a useful opportunity to measure enteropathogen transmission in populations without routine stool-based testing (*Metcalf et al., 2016*; *Arnold et al., 2018*).

These analyses had limitations. High correlation between *Salmonella* LPS Groups B and D, between ETEC LT B subunit and cholera toxin B subunit, and between norovirus GI and norovirus GII at the individual level (*Figure 2*) shows that for these pathogens seroepidemiologic analyses will be less specific than analyses based on molecular detection. *Salmonella* LPS Groups B and D have antigenic overlap in their lipopolysaccharides (*Grimont and Weill F-X, 2007*), and ETEC LT B subunit and cholera toxin B subunit are known to be immunologically cross-reactive (*Smith and Sack, 1973*). There was no known cholera transmission in the study populations, so we assumed the elevated responses to the cholera toxin B subunit reflected exposure to LT-producing ETEC; this assumption could only be confirmed with specific measures of cholera infection. Norovirus GI and GII virus-like particles are antigenically different, but cross-reactivity between norovirus genogroups is possible (*Tamminen et al., 2012*). In addition, a recent study from Uganda found high seroprevalence for both GI and GII norovirus suggesting repeated infections by viruses of the same or different genogroup could potentially boost cross-reactive antibody production (*Thorne et al., 2018*). Despite lower specificity between genogroups or serogroups in these cases, the consistency of antibody relationships with predictions based on antigenic overlap, and absence of correlation between unrelated antibodies (*Supplementary file 3*), lends support to the internal validity of the assays.

Our reliance on IgG as a sole measure of immune response did not enable finer distinction between pathogens in age-dependent acquired immunity that might be possible with additional

measures of immune function. For example, ETEC LT and *Campylobacter* IgG levels and seroprevalence consistently rose quickly by age 1 year across all of the cohorts (*Figure 3* and its supplements). We interpret the steep rise in IgG early in life to reflect higher transmission and exposure to ETEC and *Campylobacter* compared with other pathogens studied, but the steeper rise in IgG levels could also reflect a difference in immune mechanism. For example, it could reflect a more robust response to T-cell dependent antigens compared with T-cell independent antigens (e.g., *Salmonella* LPS), which might require a more mature immune system to mount a robust B-cell response (*Siegrist and Aspinall, 2009*). Yet, the steep rise in *Salmonella* LPS IgG among Haitian children (*Figure 3*), and the faster age-dependent rise in IgG response to *Salmonella* LPS compared with ETEC LT B subunit among children in the United States (*Arnold et al., 2017*) suggest that differences in IgG acquisition by age likely reflect differences in early life pathogen exposure.

Children were measured relatively infrequently in the Kenya and Haiti cohorts, which could have led to missed seroconversions if IgG levels waned below the seropositivity cutoff before a child's next measurement. Our simulations that studied the effect of more frequent sampling showed that monthly measurement could lead to higher force of infection estimates in settings with frequent exposure, and that more frequent measurements would be most valuable for highest transmission pathogens, such as ETEC and *Campylobacter* in this study (*Figure 7*). Higher resolution longitudinal measurements would also enable study of seasonal differences in enteropathogen transmission. Measurement timing in analysis populations precluded extensive evaluation of seasonal differences in IgG response, but age-adjusted seroprevalence was 1–5% higher across pathogens during the wet season in the Haiti cohort (*Supplementary file 1*). This exploratory result suggests a potential avenue of future research that could join high resolution temperature, precipitation, and antibody measurements to study seasonal drivers of enteropathogen transmission dynamics.

Another limitation is that without measures of patent infection in stool we were unable to compare multiple independent measures of transmission. Paired stool and blood specimens could help resolve some of the questions raised above related to serological assay specificity and differential immune development across pathogens. The Haiti and Tanzania cohorts were not originally designed to assess enteropathogens, and antibody measurements were not paired with measures of clinical symptoms of diarrhea or with measures of patent infection in stool. In Haiti, *Moss et al. (2014)* previously reported limited microscopy testing for protozoan cysts alongside IgG markers, but infrequent stool measurements prevented extensive comparisons. In Kenya, diarrheal stool testing during weekly follow-up visits for *Cryptosporidium* and *Giardia* showed that serology identified nearly all infections detected in stool, plus a substantial number of presumed infections not detected through diarrheal surveillance (*Supplementary file 4*); however, for other pathogens studied we did not have stool-based measures of infection. All included antigens have been characterized extensively with respect to patent infections among adults and children in other settings (details in Materials and methods). Furthermore, we observed high levels of consistency across cohorts in seroepidemiologic patterns, and consistency with general age-dependent patterns documented across diverse pathogens for which transmission begins early in life (*Arnold et al., 2017*; *Hens et al., 2012*; *Corran et al., 2007*; *Katzelnick et al., 2018*). Together, these observations suggest that the antibody dynamics and force of infection estimates from IgG responses reflect actual pathogen transmission in these cohorts, but paired stool and blood testing would provide a more definitive test.

High-resolution, longitudinal assessment of paired enteropathogen infection and antibody measurements among children could provide valuable, additional insights into the pathogen-specific antibody dynamics following primary- and secondary infections. Analyses of *Plasmodium falciparum* (*White et al., 2014*; *Helb et al., 2015*; *Rodriguez-Barraquer et al., 2018*) and dengue virus (*Salje et al., 2018*) illustrate how paired, longitudinal measurements of patent infection and antibody response enable richer characterizations of antibody dynamics and pathogen transmission. Paired longitudinal stool and antibody testing among children and adults would enable studies of age-dependent enteropathogen antibody kinetics, and if present, disentangle them from age-varying incidence rates when using serology to estimate force of infection. Studies that pair stool-based molecular testing with antibody response could also assess whether approaches to estimate enteropathogen incidence from cross-sectional samples that rely on estimates of antibody decay with time since infection (*Teunis et al., 2012*; *Simonsen et al., 2009*) could be used among children in low-resource settings. A similar approach was recently described to estimate cholera incidence among

all ages (*Azman et al., 2019*), but broader development across enteropathogens and among young children remains an open area of research.

## Conclusions

Among children in Haiti, Kenya, and Tanzania, antibody-based measures of enteropathogen infection reflected high transmission with primary exposure to most pathogens occurring by age 1–2 years. In low-resource populations, seroincidence rates and force of infection estimated beyond the age range of primary infection ideally should account for IgG boosting and waning above seropositivity cutoffs. Antibodies are a promising approach to measure population-level enteropathogen infection, and seroepidemiologic measures of heterogeneity and transmission are central considerations for their use in trials or in serologic surveillance. Our findings show that for most enteropathogens studied, the ideal window to measure heterogeneity in antibody response closes by ages 2 to 5 years in low-resource settings, and studies that plan to estimate force of infection should favor longitudinal designs with multiple measurements in this early age window.

# Materials and methods

### Key resources table

| Reagent type or resource | Designation | Source or reference | Identifiers | Additional information |
|---|---|---|---|---|
| Peptide, recombinant protein | *Giardia intestinalis* VSP-3 | PMID: 17901334 PMID: 20876825 | GenBank: XM_001707314 | Dr. Jeffrey Priest (CDC) |
| Peptide, recombinant protein | *Giardia intestinalis* VSP-5 | PMID: 11500396 PMID: 20876825 | GenBank: AF354538.1 | Dr. Jeffrey Priest (CDC) |
| Peptide, recombinant protein | *Cryptosporidium* Cp17 | PMID: 10699255 PMID: 15165066 | GenBank: AF114166 | Dr. Jeffrey Priest (CDC) |
| Peptide, recombinant protein | *Cryptosporidium* Cp23 | PMID: 8892291 PMID: 10203492 | GenBank: U34390 | Dr. Jeffrey Priest (CDC) |
| Peptide, recombinant protein | *Entamoeba histolytica* LecA | PMID: 2000392 PMID: 14741152 PMID: 24591430 | GenBank: M60498 | Dr. William Petri (University of Virginia) and Dr. Joel Herbein (TechLab) |
| Peptide, recombinant protein | *Campylobacter jejuni* p18 | PMID: 8576327 PMID: 16014430 | GenBank: X83374 | Dr. Jeffrey Priest (CDC) |
| Peptide, recombinant protein | *Campylobacter jejuni* p39 | PMID: 10688204 PMID: 16014430 | GenBank: CAL34198.1 | Dr. Jeffrey Priest (CDC) |
| Peptide, recombinant protein | ETEC heat labile toxin B subunit | PMID: 3882744 PMID: 18494692 | | Sigma-Aldrich |
| Peptide, recombinant protein | Cholera toxin B subunit | PMID: 3882744 | | Sigma-Aldrich |
| Peptide, recombinant protein | *Salmonella enterica* LPS group B | PMID: 2567429 PMID: 17329442 | | Sigma-Aldrich |
| Peptide, recombinant protein | *Salmonella enterica* LPS group D | PMID: 2567429 PMID: 17329442 | | Sigma-Aldrich |
| Peptide, recombinant protein | Norovirus GI.4 Virus Like Particles | This paper | | Dr. Jan Vinje (CDC) |
| Peptide, recombinant protein | Norovirus GII.4.NO Virus Like Particles | This paper | | Dr. Jan Vinje (CDC) |

## Ethics statement

In Haiti, the human subjects protocol was reviewed and approved by the Ethical Committee of St. Croix Hospital (Leogane, Haiti) and the institutional review board at the US Centers for Disease Control and Prevention (CDC). After listening to an overview of the study, individuals were asked for verbal consent to participate. Verbal consent was deemed appropriate by both review boards because of low literacy rates in the study population. With each longitudinal visit, the study team re-consented participants before specimen collection. Mothers provided consent for children under 7, and

children 7 years and older provided additional verbal assent. In Kenya, the human subjects protocol was reviewed and approved by institutional review boards at the Kenya Medical Research Institute (KEMRI) and at the US CDC. Primary caretakers provided written informed consent for their infant child's participation in the trial and blood specimen collection and testing (*Morris et al., 2018*). The original trial was registered at clinicaltrials.org (NCT01695304). In Tanzania, the human subjects protocol was reviewed and approved by the Institute for Medical Research Ethical Review Committee in Dar es Salaam, Tanzania and the institutional review board at the US CDC. Parents of enrolled children provided consent, and children 7 years and older also provided verbal assent before specimen collection.

## Multiplex bead assays

### Antigens

All antigens used have been well-characterized in previous studies. Lipopolysaccharides (LPS) from Group D *Salmonella enterica* serotype Enteritidis (*Strid et al., 2007*), LPS from Group B *S. enterica* serotype Typhimurium (*Strid et al., 2007*), and recombinant heat labile toxin B subunit protein (*Arnold et al., 2017*; *Levine et al., 1985*; *Flores et al., 2008*) from enterotoxigenic *Escherichia coli* (ETEC LT B subunit) were purchased from Sigma Chemical (St. Louis, MO). Recombinant *Giardia* VSP-3 and VSP-5 (*Bienz et al., 2001 Morrison et al., 2007*) and *Cryptosporidium* Cp17 and Cp23 antigens (*Perryman et al., 1996*; *Priest et al., 1999*; *Priest et al., 2000*) were expressed and purified as previously described (*Priest et al., 2010*; *Moss et al., 2004*). Recombinant *Campylobacter* p18 and p39 antigens (*Burnens et al., 1995*; *Parkhill et al., 2000*; *Schmidt-Ott et al., 2005*) were expressed and purified as previously described (*Zambrano et al., 2017*). *E. histolytica* LecA antigen (*Moss et al., 2014*; *Tannich et al., 1991*; *Houpt et al., 2004*) was kindly provided by William Petri (University of Virginia) and Joel Herbein (TechLab). Virus-like particles from norovirus GI.4 and GII.4 New Orleans were purified from a recombinant baculovirus expression system (*Arnold et al., 2017*; *Jiang et al., 1992*; *Pisanic et al., 2019*). All antibody responses were measured in multiplex bead assays on the Luminex platform at the United States Centers for Disease Control and Prevention: Haiti study (PJL's laboratory), Kenya study (JWP's laboratory), and Tanzania study (DLM's laboratory).

### Haiti

Sera from the Haiti cohort study were diluted 1:400 and analyzed by multiplex bead assay as described in detail elsewhere (*Moss et al., 2014*; *Arnold et al., 2017*; *Hamlin et al., 2012*). *Salmonella* LPS and ETEC LT B subunit antigens were coupled to SeroMap (Luminex Corp, Austin, TX) beads in buffer containing 0.85% NaCl and 10 mM $Na_2HPO_4$ at pH 7.2 (PBS) using 120 micrograms for $1.25 \times 10^7$ beads using the methods described by Moss and colleagues (*Moss et al., 2011*). Coupling conditions and externally defined cutoff values for the *Giardia*, *Cryptosporidium*, and *E. histolytica* antigens as well as for the *Schistosoma japonicum* glutathione-S-transferase (GST) negative control protein have been previously reported (*Moss et al., 2014*). Each multiplex bead assay plate (N = 25) included four control sera: one negative sample and three positive samples that spanned the response range from low to high for a selection of the antigen markers. For the positive control sample responses to the 10 enteric antigens used in this study, the average coefficient of variation (CV%) was 8.3 with a standard deviation of 4.3. The median CV% was 7.9 with a range of 2.3% to 19.9%.

### Kenya

For the Kenya study, an optimized bead coupling technique using less total protein was performed in buffer containing 0.85% NaCl and 25 mM 2-(N-morpholino)-ethanesulfonic acid at pH 5.0. The B subunit protein from cholera toxin was purchased from Sigma Chemical. The GST negative control protein (15 µg), *Cryptosporidium* Cp17 (6.8 µg) and Cp23 (12.5 µg) proteins and the *Campylobacter* p39 protein (25 µg) were coupled to $1.25 \times 10^7$ beads using the indicated protein amounts (*Zambrano et al., 2017*). The *Giardia*, *E. histolytica*, ETEC, cholera, and *Campylobacter* p18 proteins were coupled using 30 µg of protein per $1.25 \times 10^7$ beads. *Salmonella* LPS B and LPS D were coupled to the same number of beads using 60 µg and 120 µg, respectively. Blood spot elutions from the Kenya study were diluted to a final serum concentration of 1:400 (assuming 50% hematocrit) and

analyzed by multiplex bead assay as described by Morris and colleagues (*Morris et al., 2018*). Each multiplex bead assay plate (N = 11) included five control sera: one negative sample and four positive control samples offering a range of responses to various antigen markers. For the positive control sample responses to the 11 enteric antigens used in this study, the average CV% was 6.3 with a standard deviation of 3.7. The median CV% was 5.3 with a range of 1.1% to 14.9%.

### Tanzania

For the Tanzania study, the same conditions described in the Kenya study were used to couple antigens from *Giardia*, *Cryptosporidium*, *E. histolytica*, ETEC B toxin subunit, cholera B toxin subunit, GST, and *Salmonella* LPS group B and LPS group D. *Campylobacter* p39 and p18 were both coupled at 25 µg per $1.25 \times 10^7$ beads in buffer containing 0.85% NaCl and 25 mM 2-(N-morpholino)-ethanesulfonic acid at pH 5.0. Dried blood spots were eluted in the casein-based buffer described previously (*Priest et al., 2010*) and samples were diluted to either 1:400 serum dilution with 50 µl run per well for year 1, or 1:320 serum dilution with 40 µl run per well for years 2–4. The incubation steps, washes, and data collection methods used in the multiplex bead assay were performed as described previously (*Priest et al., 2010*). All samples were run in duplicate, and the average median fluorescence intensity minus background (MFI-bg) value was recorded. Each multiplex bead assay plate (N = 37) included four control sera: one negative sample and three positive control samples offering a range of responses to various antigen markers. For the positive control sample responses to the nine enteric antigens used in this study, the average CV% was 8.4 with a standard deviation of 5.3. The median CV% was 5.3 with a range of 2.6 to 15.1. The Tanzania study used different bead lots in year 1 and years 2–4; we confirmed that the use of different bead lots had no influence on the results (*Supplementary file 1*).

## Antibody distributions and determination of seropositivity

We transformed IgG levels to the $\log_{10}$ scale because the distributions were highly skewed. Means of the log-transformed data represent geometric means. We summarized the distribution of $\log_{10}$ IgG response using kernel density smoothers. In the Tanzania and Haiti cohorts, where children were measured across a broad age range, we stratified IgG distributions by each year of age <3 years to examine age-dependent changes in the population distributions. To assess potential cross-reactivity between antigens, we estimated pairwise correlations between individual-level measurements in each cohort using a Spearman rank correlation (*Zar, 2005*) and visualized the relationship for each pairwise combination with locally weighted regression fits (*Cleveland and Devlin, 1988*).

We compared three approaches to estimate seropositivity cutoffs. *Approach 1:* External known positive and negative specimens were used to determine seropositivity cutoffs for *Giardia* VSP-3 and VSP-5 antigens, *Cryptosporidium* Cp17 and Cp23 antigens, and *E. histolytica* LecA antigen. Cutoffs were determined using ROC analysis as previously described (*Moss et al., 2014*; *Morris et al., 2018*) for all antigens except for LecA, VSP-3, and VSP-5 in Haiti; in these cases, the mean plus three standard deviations of 65 specimens from citizens of the USA with no history of foreign travel were used to estimate cutoffs (*Moss et al., 2014*). *Approach 2:* We fit a 2-component, finite Gaussian mixture model (*Benaglia et al., 2009*) to the antibody distributions among children 0–1 years old, and estimated seropositivity cutoffs using the lower component's mean plus three standard deviations. The rationale for restricting the mixture model estimation in Haiti and Tanzania to children 0–1 years old was based on initial inspection of the age-stratified IgG distributions that revealed a shift from bimodal to unimodal distributions by age 3 (*Figure 1*). This approach ensured that there was a sufficiently large fraction of unexposed children in the sample to more clearly estimate a distribution among seronegative children. *Approach 3:* In the longitudinal Haiti and Kenya cohorts we identified children < 1 year old who presumably seroconverted, defined as an increase in MFI-bg values of +two or more on the $\log_{10}$ scale. A sensitivity analysis showed that an increase of 2 on the $\log_{10}$ scale was a conservative approach to identify seroconversion for most antibodies considered in this study; an increase of between 0.3 to 2.16 MFI-bg lead to optimal agreement with ROC-based and mixture model-based classifications in Kenya, and an increase of 0.92 to 2.41 led to optimal agreement across antigens and references in Haiti (*Supplementary file 5*). We then used the distribution of measurements before seroconversion to define the distribution of IgG values among the presumed unexposed. We used the mean $\log_{10}$ MFI-bg plus three standard deviations of the presumed

unexposed distribution as a seropositivity cutoff. We summarized the proportion of observations that were in agreement between the three classification approaches, and estimated Cohen's Kappa (*Cohen, 1960*). Additional details and estimates of seropositivity cutoff agreement are reported in *Supplementary file 2*. Mixture models failed to estimate realistic cutoff values if there was an insufficient number of unexposed children, which was the case for ETEC LT B subunit and cholera toxin B subunit in all cohorts, and for nearly all antigens in Tanzania where the study did not enroll children < 1 year old (*Table 1*).

In analyses of seroprevalence and seroconversion, we classified measurements as seropositive using ROC-based cutoffs if available, and mixture model-based cutoffs otherwise. There were three exceptions. By age 1 year, a majority of children across the cohorts had IgG levels near the maximum of the assay's dynamic range for ETEC LT B toxin and cholera toxin B subunit. The absence of a sufficient number of unexposed children to ETEC LT B toxin, cholera B toxin, and in some cases *Campylobacter* p18 or p39 led mixture models either to not converge or to estimate unrealistically high seropositivity cutoffs beyond the range of quantifiable levels. For these pathogens, we used seropositivity cutoffs estimated from presumed unexposed measurements in the longitudinal Haiti and Kenya cohorts (approach 3, above). High levels of agreement between classifications (*Supplementary file 2*) meant results were insensitive to choice of approach in these cohorts. We classified children as seropositive to *Giardia*, *Cryptosporidium*, *Campylobacter*, or *Salmonella* if antibody levels against either of the antigens from each pathogen were above estimated seropositivity cutoffs.

## Age-dependent antibody levels and seroprevalence curves

We estimated mean IgG levels and seroprevalence by age using semiparametric cubic splines in a generalized additive model, specifying binomial errors for seroprevalence, and random effects for children or clusters in the case of repeated observations (*Wood, 2017*; *Wood, 2012*). We also estimated the relationships by age using a stacked ensemble approach called 'super learner' that included a broader and more flexible library of machine learning algorithms (*Arnold et al., 2017*; *van der Laan et al., 2007*; *Polley et al., 2018*), and found similar fits to cubic splines. We estimated approximate, simultaneous 95% confidence intervals around the curves using a parametric bootstrap from posterior estimates of the model parameter covariance matrix (*Ruppert et al., 2003*). *Supplementary file 6* includes additional details.

## Force of infection from longitudinal data

In the Kenya and Haiti longitudinal cohorts, we estimated prospective seroconversion rates as a measure of force of infection by dividing the number of children who seroconverted by the person-time at risk between measurements. We defined incident seroconversions and seroreversions as a change in IgG across a pathogen's seropositivity cutoff. Vaccine immunogenicity and pathogen challenge studies among healthy adults often use a 4-fold increase in antibody levels (difference of +0.6 on the $\log_{10}$ scale) as a criterion for seroconversion (*Bernstein et al., 2015*; *Jin et al., 2017*; *Chakraborty et al., 2018*). In a secondary analysis aimed to capture significant changes above a pathogen's seropositivity cutoff, we defined incident boosting episodes as a $\geq 4$ fold increase in IgG to a final level above a seropositivity cutoff, and incident waning episodes as $\geq 4$ fold decrease in IgG from an initial level above a seropositivity cutoff. In the secondary definition, individuals were considered at risk for incident boosting episode if they were seronegative, if they experienced a $\geq 4$ fold increase in IgG in their first measurement period, or if they experienced a $\geq 4$ fold decrease in IgG in a preceding period (Haiti). To estimate person-time at risk used for rates and force of infection, we assumed incident changes were interval-censored and occurred at the midpoint between measurements. We estimated 95% confidence intervals for rates with 2.5 and 97.5 percentiles of a nonparametric bootstrap distribution (*Wasserman, 2004*) that resampled children with replacement to account for repeated observations.

## Force of infection from age-structured seroprevalence in Kenya

In the Kenya cohort, we estimated force of infection through age-structured seroprevalence using multiple approaches. There is a long history methods development to estimate force of infection from age-dependent seroprevalence (*Hens et al., 2012*), which is of particular interest to large-scale,

cross-sectional surveillance platforms (**Arnold et al., 2018**). Our rationale was to determine if force of infection estimates from age-structured seroprevalence were comparable to estimates from the longitudinal analysis based on incident changes in serostatus.

As we show in **Supplementary file 7**, the age dependent seroprevalence curve is the difference between the cumulative distribution functions of seroconversion times and seroreversion times. In a special case of no seroreversion, age-specific seroprevalence is thus the cumulative hazard function. The age-specific force of infection can then be estimated as the hazard of seroconverting at age $A = a$: $\lambda(a) = F'(a) / [1 - F(a)]$, where $F(a)=P(Y | A = a)$ is the proportion of the population who are seropositive at age $a$ and $F'(a)$ is the derivative of $F(a)$ with respect to $a$. Key assumptions include stationarity/homogeneity (i.e., no intervention or cohort effects) and that there is no seroreversion (**Hens et al., 2012**). There was no evidence for large changes in transmission during the studies, even due to intervention (**Supplementary file 1**). We know for many enteric pathogens children in the Kenya cohort did serorevert (e.g., **Figure 6**); when assumption is violated, estimates provide a lower-bound of a pathogen's force of infection. We considered three different estimation approaches for force of infection from age-structured seroprevalence.

## Exponential model (SIR model)

The simplest catalytic model, a susceptible-infected-recovered (SIR) model, assumed a constant force of infection over different ages, $\lambda(a) = \lambda$ and no seroreversion (**Hens et al., 2012**). In the survival analysis context, this is equivalent to assuming a constant hazard, which can be estimated with an exponential survival model. We modeled the probability of being seropositive conditional on age with a generalized linear model fit with maximum likelihood that assumed a binomial error structure and complementary log-log link (**Jewell and Laan, 1995**). We estimated average force of infection from the model's intercept term:

$$log - log[1 - P(Y = 1|A)] = log\,\lambda + \, log\,A$$

## Reversible catalytic model (SIS model)

For some infectious diseases, like malaria, reversible catalytic models have been proposed to estimate force of infection from an age-seroprevalence curve while accounting for antibody waning with time since infection (**Corran et al., 2007**). The model assumes a constant rate of seroconversion, $\lambda$, but extends the SIR model by also assuming a constant seroreversion rate, $\rho$, equivalent to a susceptible-infected-susceptible (SIS) model. We modeled the probability of a child being seropositive conditional on age as a function of these two additional parameters:

$$P(Y = 1|A) = \frac{\lambda}{\lambda + \rho}[1 - exp(-(\lambda + \rho) \cdot A)]$$

We fit the model with maximum likelihood assuming a binomial error structure and a fixed seroreversion rate. To incorporate information about the seroreversion rate in the model, we bootstrapped the dataset 1000 times, resampling children with replacement. In each bootstrap replicate, we estimated each pathogen's seroreversion rate using information from longitudinal data, and then fit the reversible catalytic model assuming a cross-sectional sample. The results are thus optimistic because they incorporate some information from the longitudinal design. Attempts to fit an SIS model assuming only a cross-sectional design with the seroreversion rate as a second free parameter led to highly unstable estimates, consistent with results presented in **Supplementary file 7** that show age-dependent seroprevalence alone does not technically contain information about seroreversion ($\rho$). As an internal validity check, we confirmed that force of infection estimates from the SIS model matched those from the SIR model for ETEC and *Campylobacter*, pathogens which had seroreversion rates that approached 0.

## Semiparametric spline model

We fit a model that allowed force of infection to vary flexibly by age using cubic splines in a generalized additive model (**Wood, 2017**). Let $\eta[P(Y = 1|A)] = logit\,P(Y = 1|A) = g(A)$ for an arbitrary function $g(\cdot)$, which we fit with cubic splines that had smoothing parameters chosen through cross-

validation. The age-specific prevalence predicted from the model is: $\hat{P}(Y=1 \mid A=a) = \frac{exp[\hat{\eta}(a)]}{1+exp[\hat{\eta}(a)]}$ and the age-specific force of infection is:

$$\hat{\lambda}(a) = \hat{\eta}'(a)\frac{exp[\hat{\eta}(a)]}{1+exp[\hat{\eta}(a)]}$$

where $\hat{\eta}'(a)$ is the first derivative of the linear predictor from the logit model (*Hens et al., 2012*). We estimated $\hat{\eta}'(a)$ and its standard error using finite differences from spline model predictions (*Wood, 2017*; *Wood, 2012*). To estimate average force of infection from the model, comparable to other methods used, for each pathogen we estimated the marginal average force of infection over the empirical age distribution in the cohort:

$$\hat{\lambda} = \int_a \hat{\lambda}(a)P(A=a)$$

We estimated approximate 95% confidence intervals for the average force of infection by simulation with a parametric bootstrap that was based on the posterior distribution of model parameters (*Ruppert et al., 2003*; *Marra and Wood, 2012*). *Supplementary file 7* provides additional details, including age-specific estimates of force of infection for each pathogen.

## Effects of sampling interval on serological force of infection estimates

We conducted a simulation study to investigate whether longer sampling intervals used in the Kenya and Haiti cohorts could lead to under-estimates of seroconversion rates. We modeled a single antigen per pathogen (e.g., Cp17 for *Cryptosporidium*). For each cohort, we created 100 simulated datasets that imputed each child's daily IgG levels. The simulation was designed provide an approximate upper bound on force of infection measured with serology. IgG levels were allowed to continuously boost and wane as long as each child's IgG trajectory remained consistent with their empirical measurements. We drew IgG boosts for each antibody from the empirical distribution of >4 fold increases in each cohort. We assumed IgG levels decayed exponentially. We estimated exponential decay parameters for each antibody using a subsample of adjacent measurements separated by <1 year with the largest decline in IgG (bottom quintile of declines), selected to reduce the possibility of intermediate exposures. IgG half-life estimates ranged from 51 to 169 days across antibodies in Haiti. In Kenya, too few children experienced reductions in IgG between measures for us to reliably estimate antibody decay rates, so the simulation assumed a fixed IgG decay rate across antibodies that corresponded to a half-life of 69 days, which was broadly consistent with the majority of IgG half-life estimates in Haiti and with 12 week half-life estimates for *Cryptosporidium* IgG levels among Canadian adults (*Priest et al., 2001*).

After simulating daily IgG trajectories for all children, we down-sampled the imputed data at intervals of 30, 90, 180, and 360 days and estimated seroconversion rates and seroreversion per the main analysis. We found that sampling intervals of 30 days adequately summarized even the most dynamic IgG trajectories. As an internal validity check, we compared seroconversion and seroreversion rates estimated from the simulations with empirical rates at comparable sampling intervals (180 days in Kenya, 360 days in Haiti) and found excellent agreement. Across the 100 simulated datasets, we quantified the potential influence of measurement frequency on force of infection estimates by estimating median rates for each pathogen and sampling interval, as well as the median difference in rates between sampling intervals. *Supplementary file 8* includes the full simulation and all details.

## Data availability and replication files

Analyses were conducted in R version 3.5.3. Data and computational notebooks used to complete the analyses are available through GitHub (*Arnold, 2019*; copy archived at https://github.com/elifesciences-publications/enterics-seroepi) and the Open Science Framework (osf.io/r4av7).

## Acknowledgements

The authors are grateful to Drs Ciara E O'Reilly and Jennifer L Murphy for oversight and field assistance with dried blood spot collection in the Kenya study, and to Katy Hamlin for assistance with the

analysis of specimens in the Haiti study. We thank William Petri and Joel Herbein for the kind gift of LecA antigen.

## Additional information

### Funding

| Funder | Grant reference number | Author |
|---|---|---|
| National Institutes of Health | K01-AI119180 | Benjamin F Arnold |
| Bill and Melinda Gates Foundation | OPP1022543 | Patrick J Lammie |

The funders had no role in study design, data collection and interpretation, or the decision to submit the work for publication.

### Author contributions

Benjamin F Arnold, Conceptualization, Data curation, Software, Formal analysis, Funding acquisition, Investigation, Visualization, Methodology, Writing—original draft, Project administration, Writing—review and editing; Diana L Martin, Resources, Data curation, Supervision, Validation, Writing—original draft, Project administration, Writing—review and editing; Jane Juma, Jan Vinjé, Resources, Validation, Writing—review and editing; Harran Mkocha, Richard Omore, Resources, Validation, Project administration, Writing—review and editing; John B Ochieng, Supervision, Project administration, Writing—review and editing; Gretchen M Cooley, Resources, Supervision, Validation, Investigation, Writing—review and editing; E Brook Goodhew, Veronica Costantini, Resources, Validation, Investigation, Writing—review and editing; Jamae F Morris, Resources, Data curation, Supervision, Investigation, Project administration, Writing—review and editing; Patrick J Lammie, Conceptualization, Supervision, Funding acquisition, Methodology, Writing—review and editing; Jeffrey W Priest, Conceptualization, Resources, Data curation, Supervision, Validation, Investigation, Methodology, Writing—original draft, Writing—review and editing

### Author ORCIDs

Benjamin F Arnold (iD) https://orcid.org/0000-0001-6105-7295

### Ethics

Human subjects: In Haiti, the human subjects protocol was reviewed and approved by the Ethical Committee of St Croix Hospital (Leogane, Haiti) and the institutional review board at the US Centers for Disease Control and Prevention (CDC). After listening to an overview of the study, individuals were asked for verbal consent to participate. Verbal consent was deemed appropriate by both review boards because of low literacy rates in the study population. With each longitudinal visit, the study team re-consented participants before specimen collection. Mothers provided consent for children under 7, and children 7 years and older provided additional verbal assent. In Kenya, the human subjects protocol was reviewed and approved by institutional review boards at the Kenya Medical Research Institute (KEMRI) and at the US CDC. Primary caretakers provided written informed consent for their infant child's participation in the trial and blood specimen collection and testing. The original trial was registered at clinicaltrials.org (NCT01695304). In Tanzania, the human subjects protocol was reviewed and approved by the Institute for Medical Research Ethical Review Committee in Dar es Salaam, Tanzania, and the institutional review board at the US CDC. Parents of enrolled children provided consent, and children 7 years and older also provided verbal assent before specimen collection.

### Decision letter and Author response

Decision letter https://doi.org/10.7554/eLife.45594.033
Author response https://doi.org/10.7554/eLife.45594.034

# Additional files

## Supplementary files

• Supplementary file 1. Effect of intervention, bead lot, and season on enteropathogen antibody response (osf.io/6br2f).
DOI: https://doi.org/10.7554/eLife.45594.021

• Supplementary file 2. Classification agreement between different seropositivity cutoff approaches (osf.io/7x6sw).
DOI: https://doi.org/10.7554/eLife.45594.022

• Supplementary file 3. Joint distributions of antibody response (osf.io/wchzq).
DOI: https://doi.org/10.7554/eLife.45594.023

• Supplementary file 4. IgG measurements in the Kenya cohort among children with- and without confirmed *Cryptosporidium* and *Giardia* infections in diarrheal stools (osf.io/e4tbg).
DOI: https://doi.org/10.7554/eLife.45594.024

• Supplementary file 5. Sensitivity analyses: fold-changes in IgG used to identify presumed unexposed measurements and force of infection in Haiti and Kenya (osf.io/u79bm).
DOI: https://doi.org/10.7554/eLife.45594.025

• Supplementary file 6. Estimation of age-dependent means and seroprevalence using multiple approaches (osf.io/r25hp).
DOI: https://doi.org/10.7554/eLife.45594.026

• Supplementary file 7. Estimation of force of infection from age-structured seroprevalence in Kenya (osf.io/9wbh5).
DOI: https://doi.org/10.7554/eLife.45594.027

• Supplementary file 8. Simulation study to assess the influence of sampling intervals on serological estimates of force of infection (osf.io/9zt4d).
DOI: https://doi.org/10.7554/eLife.45594.028

• Transparent reporting form
DOI: https://doi.org/10.7554/eLife.45594.029

## Data availability

Analyses were conducted in R version 3.5.3. Data and computational notebooks used to complete the analyses are available through GitHub (https://github.com/ben-arnold/enterics-seroepi; copy archived at https://github.com/elifesciences-publications/enterics-seroepi) and the Open Science Framework (osf.io/r4av7).

The following dataset was generated:

| Author(s) | Year | Dataset title | Dataset URL | Database and Identifier |
|---|---|---|---|---|
| Arnold BF, Martin DL, Juma J, Mkocha H, Ochieng JB, Cooley GM, Richard Omore R, Goodhew EB, Morris JF, Costantini V, Vinjé J, Lammie PJ, Priest JW | 2019 | Data and computational notebooks used to complete the analyses in Enteropathogen antibody dynamics and force of infection among children in low-resource settings | https://doi.org/10.17605/osf.io/r4av7 | The Open Science Framework, 10.17605/osf.io/r4av7 |

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
