## [Decision Letter]

Thank you for submitting your article "Enteropathogen antibody dynamics and force of infection among children in low-resource settings" for consideration by *eLife*. Your article has been reviewed by three peer reviewers, and the evaluation has been overseen by a Reviewing Editor and Neil Ferguson as the Senior Editor. The following individuals involved in review of your submission have agreed to reveal their identity: Michael White (Reviewer #1); Daniel T Leung (Reviewer #2); Andrew Azman (Reviewer #3).

The reviewers have discussed the reviews with one another and the Reviewing Editor has drafted this decision to help you prepare a revised submission.

Summary:

This manuscript reports on the dynamics of enteropathogen-specific antibody responses in three cohorts of children in LMICs. While largely descriptive in nature, this manuscript provides an important foundational look at enteropathogen antibody kinetics and points towards important methodological issues that must be solved in the future if we are to improve the use of serology for inference about enteric infection incidence. Hence this is an important and innovative piece of work that pushes the boundaries in integrated serological surveillance, beyond the past work in the field by yourselves and other groups. It opens up the potential for use of serosurveillance for enteric infections, to measure impact of public health interventions, and to better understand disease transmission. Despite some concerns, this report has the potential to be highly influential in the field of enteric disease epidemiology.

Essential revisions:

1) The conclusions of the paper are based only on analysis of serologic data, without demonstrated exposure confirmed by molecular or microbial detection, and longitudinal measures are 6-12 months apart. Thus, the poor granularity of data points may result in misclassifications of "primary" infection, and "boosting" due to repeated exposures. For example, in longitudinal studies of cholera patients, approximately 30% have a 4-fold "boosting" of antibody responses in the 6-12 months following an initial infection. More frequent exposures may not be well-captured by the yearly sampling that was done in Haiti, or even the 6 months apart samples from Kenya. We would like to see more exploration of this, and if possible, some method for correcting/adjusting this. For instance, the discussion notes that for two of the three studies, there were no paired stool samples. However, for the Kenyan study by Morris et al., this data was available. Although the number of PCR positive stool samples was small, it would still be incredibly valuable to look at antibody responses in PCR +ve versus PCR -ve samples. This could be done, for example, in Supplementary file 1.

2) With such dynamic post-infection kinetics the idea of seroprevalence more complicated than traditional settings where it is more or less a saturating state. In these cases, as the authors note in the discussion, the measured antibody response reflects both the effect of cumulative exposures and the time since one's last infection. It would be helpful for the authors to expand more on what seroprevalence actually means in this context and whether cut-offs should be age dependent or whether we should abandon this concept in adults?

3) It would be helpful if the authors could briefly discuss how they may disentangle age-specific serologic kinetics from age-specific incidence rates (perhaps due to different exposures/mixing patterns with age). This will certainly complicate interpretation of some epidemiologic inference.

4) It would be helpful for the authors to expand on the suggested age range where seroprevalence data (age-specific or aggregate) may be the most informative. I wonder if some measure of entropy could help.

5) The biggest serologic rises in early childhood is seen in T-cell-dependent antigens (ETEC LTB, campylobacter p18/39), and less so on T-independent antigens (*Salmonella* LPS). This could be because the low "seroconversions" in *Salmonella* LPS IgG in early childhood are a reflection of an immature immune system (we know that polysaccharide-specific IgG2 also increases with age), rather than a relative lack of exposures. A discussion of this is warranted at least, and if data sets are available, examination of longitudinal and cross-sectional enteric pathogen IgG responses from cohorts in higher-income countries (where primary enteropathogen exposures occur later in childhood due to improved WASH or rota vaccination) would help to tease out the specificity of such findings to this age group in LMICs.

6) There is no mention of seasonality in either the description of sampling methodology or in the analysis. We know that several enteropathogens (ETEC and norovirus in particular have been well-studied) occur with seasonal variation – this needs to be accounted for in the analysis.

7) Seroconversion is defined as 4-fold; however, in methods for seropositivity cutoffs, seroconversion was defined as increase in MFI-bg values of +2 or more on log10 scale (subsection “Antibody distributions and determination of seropositivity”). Though the 4-fold definition has been used by other groups, it is quite arbitrary and sensitivity analyses should be performed with other definitions of seroconversion.

8) The authors note that the slope of the relationship between seroprevalence and foi differs between Kenya and Haiti and suggest it may be due to the older ages included in the Kenya cohort. The authors could explore whether this is likely the case by looking at how the slope changes when analysis is restricted to lower age groups. Sample size may be a challenge but it seems like this may strengthen their explanation.

9) The authors note that the confidence intervals using the RCM model are small. These confidence intervals do not take into account the seroreversion parameter and it seems like this could be jointly estimated from longitudinal data and the age-specific seroprevalence curves.

---

## [Author Response]

Essential revisions:1) The conclusions of the paper are based only on analysis of serologic data, without demonstrated exposure confirmed by molecular or microbial detection, and longitudinal measures are 6-12 months apart. Thus, the poor granularity of data points may result in misclassifications of "primary" infection, and "boosting" due to repeated exposures. For example, in longitudinal studies of cholera patients, approximately 30% have a 4-fold "boosting" of antibody responses in the 6-12 months following an initial infection. More frequent exposures may not be well-captured by the yearly sampling that was done in Haiti, or even the 6 months apart samples from Kenya. We would like to see more exploration of this, and if possible, some method for correcting/adjusting this. For instance, the discussion notes that for two of the three studies, there were no paired stool samples. However, for the Kenyan study by Morris et al., this data was available. Although the number of PCR positive stool samples was small, it would still be incredibly valuable to look at antibody responses in PCR +ve versus PCR -ve samples. This could be done, for example, in Supplementary file 1.

We agree that measurement frequency in the longitudinal cohorts (6 months in Kenya, 12 months in Haiti) combined with absence of paired stool-based testing could result in misclassification of “primary infection” and “boosting” of antibody responses following an initial infection. We would expect misclassification arising from the coarsened serological measurements to lead to the estimates presented to represent lower bounds on the force of infection because we might under-estimate the incidence of actual infection. We have explored this issue in more detail through two additions to the paper.

First, we conducted a simulation study in the Haiti and Kenya cohorts that imputed each child’s IgG levels with daily resolution, with IgG kinetics modeled from empirical measures of boosting levels and waning rates in the cohorts. Simulation results suggest that, as predicted, less frequent monitoring could under-estimate seroconversion rates in settings where children have very frequent repeated infections. The influence of measurement frequency on estimates was relatively small in magnitude for most pathogens (+0.1 to +0.9 additional seroconversions per child-year detected with monthly monitoring compared with measurements every 6- or 12-months), but was more significant for highest transmission pathogens in this study, ETEC and *Campylobacter.* The new text added to the results, included below, summarizes the main findings. Supplementary file 8 describes the simulation in detail, with a summary of the approach in the Materials and methods. A new Figure 7 summarizes the main results.

New text added to Results:

“We conducted a simulation study to investigate whether longer sampling intervals in the cohorts (6 months in Kenya, 12 months in Haiti) could lead us to miss more frequent exposures and thus under-estimate force of infection. […]However, for pathogens with highest seroconversion rates, ETEC and *Campylobacter*, increases in rates estimated with 30-day sampling intervals detected a median of 4 to 8 additional seroconversions per child-year at risk compared with empirical rates (Figure 7).”

Second, we completed an analysis in the Kenya cohort that compared IgG responses among children who had confirmed *Cryptosporidium* and *Giardia* infections in diarrheal stools during weekly longitudinal monitoring, versus those who did not. *Cryptosporidium* and *Giardia* were the only two pathogens studied with serology that were also tested in stool. A new Supplementary file 4 includes the full analysis. We added the following text to the Results:

“Comparison of serology with stool-based measures of infection

The Kenya study monitored diarrhea symptoms in weekly visits between enrollment and follow-up. […] This suggests that many children were not shedding oocysts at the time of diarrheal stool collection, or many infections with these two pathogens were asymptomatic. Supplementary file 4 includes additional details.”

And to the Limitations section of the Discussion:

“In Kenya, diarrheal stool testing during weekly follow-up visits for Cryptosporidium and Giardia showed that serology identified nearly all infections detected in stool, plus a substantial number of presumed infections not detected through diarrheal surveillance (Supplementary file 4); however, for other pathogens studied we did not have stool-based measures of infection.”

2) With such dynamic post-infection kinetics the idea of seroprevalence more complicated than traditional settings where it is more or less a saturating state. In these cases, as the authors note in the discussion, the measured antibody response reflects both the effect of cumulative exposures and the time since one's last infection. It would be helpful for the authors to expand more on what seroprevalence actually means in this context and whether cut-offs should be age dependent or whether we should abandon this concept in adults?

We expanded on interpretation of seroprevalence in the Discussion to provide additional thoughts related to cutoff interpretation and use (text below). We hesitate to extrapolate our findings from this study to comment on the appropriateness of age-dependent cutoff values or the concept of seropositivity among adults because little is known about how durable the IgG response is for most of these pathogens outside of studies among adults in the United States and Europe, where transmission is much less intense. Among children we demonstrated a consistent shift at the population level from bi-modal distributions to unimodal distributions for antibody responses to pathogens such as *Giardia*, but it is unclear whether the population distribution stabilizes or whether it continues to shift lower as individuals age, gradually moving back to more closely approximating the “unexposed” distribution. As two of the reviewers recently demonstrated for cholera IgG among people of all-ages in Bangladesh (Azman et al., 2019), for cholera in that setting seropositivity cutoffs appear to remain useful even among adults because CTB IgG responses decay sufficiently over approximately 1-2 years to enable detection of boosting due to new infections.

Revised Discussion section:

“The shift of IgG distributions from bimodal to unimodal for many pathogens (*Giardia, Cryptosporidium, E. histolytica*, and *Campylobacter*), resulting from a combination of antibody boosting, waning and adaptive acquired immunity, complicates the interpretation of seropositivity at older ages: among older children a seronegative response could either mean the children were never exposed or they were previously exposed but antibody levels waned below seropositivity cutoffs. […] Increases in mean IgG levels and seroprevalence with age imply IgG boosting from new infections or repeated infections outpaced IgG decay for all enteropathogens studied until at least age 3 years, and for many pathogens through age 10 years; seroprevalence thus reflects a conservative lower bound of a population’s cumulative exposure over this age range.

The age range over which seroprevalence provided useful epidemiologic information varied by pathogen and cohort. In Haiti, 100% of children were seropositive to ETEC LT B toxin before age 12 months, though seroprevalence did not exceed 90% for most other pathogens until age 5 years in Haiti (Figure 3). In Kenya, the age range of 4-18 months captured wide variation in seroprevalence for most pathogens, and in Tanzania inclusion of children ages 1 and older missed the key window of variation in antibody response for all pathogens except E. histolytica and *Salmonella* (Figure 3—figure supplements 1 and 2). Studies that extend beyond 10 years into adolescence and adulthood would help determine whether enteropathogen seroprevalence remains sufficiently high that it no longer provides useful epidemiologic information. The shift in IgG distributions for some pathogens results raises the question of whether population mean IgG levels stabilize at a new “set point” with repeated infections as has been observed for dengue serotypes (40); if so, then the use of fold-changes in IgG would be preferred to seropositivity cutoffs to identify incident infections among older ages.

In low-resource settings, measuring a sufficient number of young children before primary infection, preferably with longitudinal measurements, will help ensure that within-sample seropositivity cutoff estimation is possible. Two-component mixture models fit the data and provided reasonable cutoff estimates only when restricted to an age range that included clearly delineated subpopulations of seronegative and seropositive responses. For most pathogens studied, this required measurements among children <1 year old, an age range during which IgG responses still followed a bimodal distribution. The only reliable approach to estimate seropositivity cutoffs for the highest transmission pathogens like ETEC and Campylobacter was to estimate a distribution among presumed unexposed by identifying measurements among children who subsequently experienced a large increase in IgG, a strategy only possible in a longitudinal design. Although not attempted here, studies that wish to compare antibody response and seroprevalence between different sites should use a common assay platform and materials (e.g., shared bead coupling) with jointly estimated seropositivity cutoffs to help ensure comparability across sites, since there are currently no global reference standards to translate arbitrary units into antibody titers for enteropathogens.”

3) It would be helpful if the authors could briefly discuss how they may disentangle age-specific serologic kinetics from age-specific incidence rates (perhaps due to different exposures/mixing patterns with age). This will certainly complicate interpretation of some epidemiologic inference.

We agree that if age-specific differences in antibody kinetics exist, that it could complicate epidemiologic inference in a setting with age-varying incidence. Our view is that longitudinal designs that pair high resolution measures of infection in stool with antibody measurements would be essential to disentangle the two processes. We added the following sentence to the Discussion:

“Paired longitudinal stool and antibody testing among children and adults would enable studies of age-dependent enteropathogen antibody kinetics, and if present, disentangle them from age-varying incidence rates when using serology to estimate force of infection.”

4) It would be helpful for the authors to expand on the suggested age range where seroprevalence data (age-specific or aggregate) may be the most informative. I wonder if some measure of entropy could help.

Please see our additions in response to comment 2.

5) The biggest serologic rises in early childhood is seen in T-cell-dependent antigens (ETEC LTB, campylobacter p18/39), and less so on T-independent antigens (Salmonella LPS). This could be because the low "seroconversions" in Salmonella LPS IgG in early childhood are a reflection of an immature immune system (we know that polysaccharide-specific IgG2 also increases with age), rather than a relative lack of exposures. A discussion of this is warranted at least, and if data sets are available, examination of longitudinal and cross-sectional enteric pathogen IgG responses from cohorts in higher-income countries (where primary enteropathogen exposures occur later in childhood due to improved WASH or rota vaccination) would help to tease out the specificity of such findings to this age group in LMICs.

Thank you very much for providing this insightful hypothesis. Based on this, we added text to the Discussion to mention this possibility and discuss the idea with respect to age-dependent IgG responses measured among children in the United States, which we published previously.

“Our reliance on IgG as a sole measure of immune response did not enable finer distinction between pathogens in age-dependent acquired immunity that might be possible with additional measures of immune function. […] Yet, the steep rise in *Salmonella* LPS IgG among Haitian children (Figure 3), and the faster age-dependent rise in IgG response to *Salmonella* LPS compared with ETEC LT B subunit among children in the United States (15) suggest that differences in IgG acquisition by age likely reflect differences in early life pathogen exposure.”

6) There is no mention of seasonality in either the description of sampling methodology or in the analysis. We know that several enteropathogens (ETEC and norovirus in particular have been well-studied) occur with seasonal variation – this needs to be accounted for in the analysis.

We have added more specific details about sample timing with respect to seasonality (additions summarized below). In the Kenya and Tanzania studies, specimen collection took place during narrow windows that did not capture seasonal variation. In Kenya, all specimens were collected in two, three-week windows: between Jan 30 – Feb 22, 2013 at enrollment, and Aug 12 – Sep 4, 2013 at follow-up. In western Kenya, the there are two periods of seasonally heavy rain: March – May and November – December. Thus, specimen collection took place during dry periods and the follow-up period captured the March – May wet season. The Tanzania study collected specimens between October and December of each year (2012, 2013, 2014, 2015), just before the beginning of the rainy season.

In Haiti, there are two annual rainy seasons: April–June and August–October. The study collected measurements in both dry and wet seasons. A previous study of the protozoan IgG responses measured in this cohort found slightly higher mean levels of *Giardia* VSP and *E. histolytica* LecA antigens (Moss et al., 2014). We extended the previous analysis to include all antigens in the present study and to examine both geometric mean IgG levels and seroprevalence by season. We examined antibody response by season and found some evidence for higher mean antibody levels and seroprevalence across all pathogens during the wet season. However, the differences were small in magnitude (e.g., increases in seroprevalence of 1 to 4 percentage points during the wet season, depending on the pathogen; prevalence ratios ranged from 1.01 to 1.07). We examined age-dependent patterns in IgG levels and found slightly higher levels during the wet season but no substantive difference in the shape of the curves. We summarized the results in Supplementary file 1. Based on the results, we concluded that season was not a major determinant of the main results described in the Haitian cohort. We have mentioned this dimension to the study and future work in the Discussion.

“Higher resolution longitudinal measurements would also enable study of seasonal differences in enteropathogen transmission. Measurement timing in the analysis populations precluded extensive evaluation of seasonal differences in IgG response, but age-adjusted seroprevalence was 1-5% higher across pathogens during the wet season in the Haiti cohort (Supplementary file 1). This exploratory result suggests a potential avenue of future research that could join high resolution temperature, precipitation, and antibody measurements to study seasonal drivers of enteropathogen transmission dynamics.”

7) Seroconversion is defined as 4-fold; however, in methods for seropositivity cutoffs, seroconversion was defined as increase in MFI-bg values of +2 or more on log10 scale (subsection “Antibody distributions and determination of seropositivity”). Though the 4-fold definition has been used by other groups, it is quite arbitrary and sensitivity analyses should be performed with other definitions of seroconversion.

Thank you for this suggestion. We agree that the use of a 4-fold increase is somewhat arbitrary in this context, though there is some justification of that magnitude of IgG increase to define seroconversion based on enteric pathogen challenge studies (e.g., Moe et al., 2004 for norovirus https://www.ncbi.nlm.nih.gov/pubmed/15539501). Our use of +2 on the log_10_ scale (100-fold increase) to identify “presumed unexposed” children was intentionally conservative to identify measurements that were highly likely to have been among unexposed.

In response to this suggestion, we conducted sensitivity analyses in Haitian and Kenyan cohorts to examine the sensitivity of the results to the choice of IgG fold-change used to define boosting and waning. We varied the fold-change in IgG from 2 to 10 and repeated boosting and waning incidence rate analyses.

In Haiti, larger values resulted in lower rates, but the rates typically declined non-linearly and, interestingly, converged with rates based on seropositivity around a 4-fold increase. Incident boosting rates using a 4-fold increase were very similar to use of seroconversion rates based on crossing seropositivity cutoffs, and were statistically indistinguishable for all pathogens except for *Cryptosporidium*, where rates did not converge until using a 5-6 fold increase in IgG. In Kenya, estimates were insensitive to the choice of fold-change in IgG used to define incident boosting, and estimates were near identical to seroconversion rates across all fold-changes in IgG. This was presumably because most increases in IgG were large in the Kenyan cohort, reflecting a primary infection for most children in this early age range. Supplementary file 5 includes this new sensitivity analysis. We added the following text to the manuscript:

“Sensitivity analyses that defined incident boosting over a range of 2-fold to 10-fold increases in IgG showed force of infection estimates were relatively stable across a wide range of definitions. In Haiti, the only pathogen for which force of infection estimated using a 4-fold increase in IgG was significantly higher than the seroconversion rate was Cryptosporidium (Supplementary file 5).”

8) The authors note that the slope of the relationship between seroprevalence and foi differs between Kenya and Haiti and suggest it may be due to the older ages included in the Kenya cohort. The authors could explore whether this is likely the case by looking at how the slope changes when analysis is restricted to lower age groups. Sample size may be a challenge but it seems like this may strengthen their explanation.

Thank you for this suggestion. On your recommendation, we conducted a sensitivity analysis whereby we progressively restricted the age range of the Haiti cohort with progressively narrow bands from 0–5 years to 0–2 years. We stopped at 0–2 years because there were fewer than 100 measurements among ages 0-1 years. The result was consistent with our earlier conjecture: the relationship was steepest amongst the youngest age band and gradually flattened as measurements among older children were included in the analysis. We added this analysis as a new supplement to Figure 5. We added the following text to the manuscript:

“Consistent with this interpretation, when we progressively narrowed the age range of the Haitian cohort and repeated the analysis, the relationship was steeper when estimated among children ages 0–2 years and flattened as measurements among older children were added (Figure 5—figure supplement 1).”

9) The authors note that the confidence intervals using the RCM model are small. These confidence intervals do not take into account the seroreversion parameter and it seems like this could be jointly estimated from longitudinal data and the age-specific seroprevalence curves.

Thank you for this suggestion. We modified the modeling approach slightly to include uncertainty in the seroreversion parameter in the reversible catalytic model. We elected to estimate the confidence intervals using a non-parametric bootstrap, and within each bootstrap replicate we re-estimated the seroreversion rate using the longitudinal information in the replicate sample. This led to slightly, but not substantially, wider confidence intervals on the seroconversion rate compared with the asymptotic intervals that we originally presented. Figure 6 now includes the revised confidence intervals, and the direct comparison of confidence interval width is presented in Supplementary file 7 that generates the figure.

Revised text in Materials and methods:

“We fit the model with maximum likelihood assuming a binomial error structure and a fixed seroreversion rate. […] Attempts to fit an SIR model assuming only a cross-sectional design with the seroreversion rate as a second free parameter led to highly unstable estimates, consistent with results presented in Supplementary file 7 that show age-dependent seroprevalence alone does not technically contain information about seroreversion (ρ).”